# Undersampling is a Minimax Optimal Robustness Intervention in Nonparametric Classification

**Niladri S. Chatterji**\*  *niladri@cs.stanford.edu*
*Department of Computer Science*
*Stanford University*

**Saminul Haque**\*  *saminulh@stanford.edu*
*Department of Computer Science*
*Stanford University*

**Tatsunori B. Hashimoto**  *thashim@stanford.edu*
*Department of Computer Science*
*Stanford University*

**Reviewed on OpenReview:** *https://openreview.net/forum?id=r6oHDYOZ6p*

## Abstract

While a broad range of techniques have been proposed to tackle distribution shift, the simple baseline of training on an *undersampled* balanced dataset often achieves close to state-of-the-art-accuracy across several popular benchmarks. This is rather surprising, since undersampling algorithms discard excess majority group data. To understand this phenomenon, we ask if learning is fundamentally constrained by a lack of minority group samples. We prove that this is indeed the case in the setting of nonparametric binary classification. Our results show that in the worst case, an algorithm cannot outperform undersampling unless there is a high degree of overlap between the train and test distributions (which is unlikely to be the case in real-world datasets), or if the algorithm leverages additional structure about the distribution shift. In particular, in the case of label shift we show that there is always an undersampling algorithm that is minimax optimal. In the case of group-covariate shift we show that there is an undersampling algorithm that is minimax optimal when the overlap between the group distributions is small. We also perform an experimental case study on a label shift dataset and find that in line with our theory, the test accuracy of robust neural network classifiers is constrained by the number of minority samples.

## 1 Introduction

A key challenge facing the machine learning community is to design models that are robust to distribution shift. When there is a mismatch between the train and test distributions, current models are often brittle and perform poorly on rare examples (Hovy & Søgaard, 2015; Blodgett et al., 2016; Tatman, 2017; Hashimoto et al., 2018; Alcorn et al., 2019). In this paper, our focus is on group-structured distribution shifts. In the training set, we have many samples from a *majority* group and relatively few samples from the *minority* group, while during test time we are equally likely to get a sample from either group.

To tackle such distribution shifts, a naïve algorithm is one that first *undersamples* the training data by discarding excess majority group samples (Kubat & Matwin, 1997; Wallace et al., 2011) and then trains a model on this resulting dataset (see Figure 1 for an illustration of this algorithm). The samples that remain in this undersampled dataset constitute i.i.d. draws from the test distribution. Therefore, while a classifier trained on this pruned dataset cannot suffer biases due to distribution shift, this algorithm is clearly wasteful, as it discards training samples. This perceived inefficiency of undersampling has led to the design of several algorithms to combat such distribution shift (Chawla et al., 2002; Lipton et al., 2018;

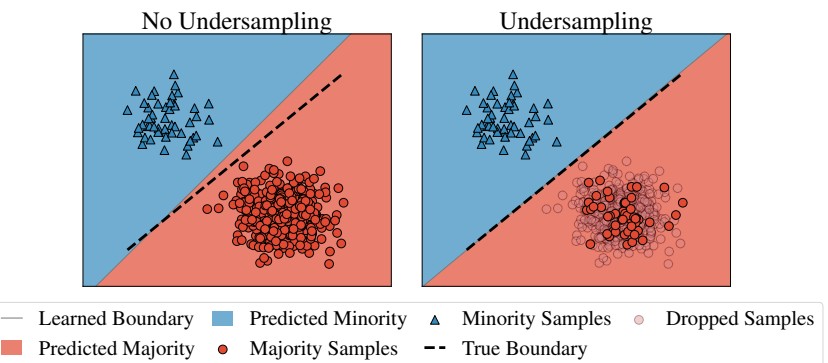

Figure 1: Example with linear models and linearly separable data. On the left we have the maximum margin classifier over the entire dataset, and on the right we have the maximum margin classifier over the undersampled dataset. The undersampled classifier is less biased and aligns more closely with the true boundary.

Sagawa et al., 2020; Cao et al., 2019; Menon et al., 2020; Ye et al., 2020; Kini et al., 2021; Wang et al., 2022). In spite of this algorithmic progress, the simple baseline of training models on an undersampled dataset remains competitive. In the case of label shift, where one class label is overrepresented in the training data, this has been observed by Cui et al. (2019); Cao et al. (2019), and Yang & Xu (2020). While in the case of group-covariate shift, a study by Idrissi et al. (2022) showed that the empirical effectiveness of these more complicated algorithms is limited.

For example, Idrissi et al. (2022) showed that on the group-covariate shift CelebA dataset the worst-group accuracy of a ResNet-50 model on the undersampled CelebA dataset which *discards 97%* of the available training data is as good as methods that use all of available data such as importance-weighted ERM (Shimodaira, 2000), Group-DRO (Sagawa et al., 2020) and Just-Train-Twice (Liu et al., 2021). In Table 1, we report the performance of the undersampled classifier compared to the state-of-the-art-methods in the literature across several label shift and group-covariate shift datasets. We find that, although undersampling isn't always the optimal robustness algorithm, it is typically a very competitive baseline and within 1–4% the performance of the best method.

Table 1: Performance of undersampled classifier compared to the best classifier across several popular label shift and group-covariate shift datasets. When reporting worst-group accuracy we denote it by a ⋆. When available, we report the 95% confidence interval. We find that the undersampled classifier is always within 1–4% of the best performing robustness algorithm, except on the CIFAR100 and MultiNLI datasets. In Appendix F we provide more details about each of the results in the table.

| Shift Type | Dataset/Paper | Test/Worst-Group⋆ Accuracy | |
| --- | --- | --- | --- |
| | | Best | Undersampled |
| Label | Imb. CIFAR10 (step 10) (Cao et al., 2019) | 87.81 | 84.59 |
| | Imb. CIFAR100 (step 10) (Cao et al., 2019) | 59.46 | 53.08 |
| Group-Covariate | CelebA (Idrissi et al., 2022) | $86.9 \pm 1.1^\star$ | $85.6 \pm 2.3^\star$ |
| | Waterbirds (Idrissi et al., 2022) | $87.6 \pm 1.6^\star$ | $89.1 \pm 1.1^\star$ |
| | MultiNLI (Idrissi et al., 2022) | $78.0 \pm 0.7^\star$ | $68.9 \pm 0.8^\star$ |
| | CivilComments (Idrissi et al., 2022) | $72.0 \pm 1.9^\star$ | $71.8 \pm 1.4^\star$ |

Inspired by the strong performance of undersampling in these experiments, we ask:

*Is the performance of a model under distribution shift fundamentally
constrained by the lack of minority group samples?*

To answer this question we analyze the *minimax excess risk*. We lower bound the minimax excess risk to prove that the performance of *any* algorithm is lower bounded only as a function of the minority samples ($n_{\mathsf{min}}$). This shows that even if a robust algorithm optimally trades off between the bias and the variance, it is fundamentally constrained by the variance on the minority group which decreases only with $n_{\mathsf{min}}$.

**Our contributions.** In our paper, we consider the well-studied setting of nonparametric binary classification (Tsybakov, 2010). By operating in this nonparametric regime we are able to study the properties of undersampling in rich data distributions, but are able to circumvent the complications that arise due to the optimization and implicit bias of parametric models.

We provide insights into this question in the label shift scenario, where one of the labels is overrepresented in the training data, $\mathsf{P}_{\mathsf{train}}(y = 1) \geq \mathsf{P}_{\mathsf{train}}(y = -1)$, whereas the test samples are equally likely to come from either class. Here the class-conditional distribution $\mathsf{P}(x \mid y)$ is Lipschitz in $x$. We show that in the label shift setting there is a fundamental constraint, and that the minimax excess risk of *any robust learning method* is lower bounded by $1/n_{\mathsf{min}}^{1/3}$. That is, minority group samples fundamentally constrain performance under distribution shift. Furthermore, by leveraging previous results about nonparametric density estimation (Freedman & Diaconis, 1981) we show a matching upper bound on the excess risk of a standard binning estimator trained on an undersampled dataset to demonstrate that undersampling is optimal.

Further, we experimentally show in a label shift dataset (Imbalanced Binary CIFAR10) that the accuracy of popular classifiers generally follow the trends predicted by our theory. When the minority samples are increased, the accuracy of these classifiers increases drastically, whereas when the number of majority samples are increased the gains in the accuracy are marginal at best.

We also study the covariate shift case. In this setting, there has been extensive work studying the effectiveness of transfer (Kpotufe & Martinet, 2018; Hanneke & Kpotufe, 2019) from train to test distributions, often focusing on deriving specific conditions under which this transfer is possible. In this work, we demonstrate that when the overlap (defined in terms of total variation distance) between the group distributions $\mathsf{P}_a$ and $\mathsf{P}_b$ is small, transfer is difficult, and that the minimax excess risk of any robust learning algorithm is lower bounded by $1/n_{\mathsf{min}}^{1/3}$. While this prior work also shows the impossibility of using majority group samples in the extreme case with no overlap, our results provide a simple lower bound that shows that the amount of overlap needed to make transfer feasible is unrealistic. We also show that this lower bound is tight, by proving an upper bound on the excess risk of the binning estimator acting on the undersampled dataset.

Taken together, our results underline the need to move beyond designing "general-purpose" robustness algorithms (like importance-weighting (Cao et al., 2019; Menon et al., 2020; Kini et al., 2021; Wang et al., 2022), g-DRO (Sagawa et al., 2020), JTT (Liu et al., 2021), SMOTE (Chawla et al., 2002), etc.) that are agnostic to the structure in the distribution shift. Our worst case analysis highlights that to successfully beat undersampling, an algorithm must leverage additional structure in the distribution shift.

## 2 Related work

On several group-covariate shift benchmarks (CelebA, CivilComments, Waterbirds), Idrissi et al. (2022) showed that training ResNet classifiers on an undersampled dataset either outperforms or performs as well as other popular reweighting methods like Group-DRO (Sagawa et al., 2020), reweighted ERM, and Just-Train-Twice (Liu et al., 2021). They find Group-DRO performs comparably to undersampling, while both tend to outperform methods that don't utilize group information.

One classic method to tackle distribution shift is importance weighting (Shimodaira, 2000), which reweights the loss of the minority group samples to yield an unbiased estimate of the loss. However, recent work (Byrd & Lipton, 2019; Xu et al., 2020) has demonstrated the ineffectiveness of such methods when applied to overparameterized neural networks. Many followup papers (Cao et al., 2019; Ye et al., 2020; Menon et al.,

2020; Kini et al., 2021; Wang et al., 2022) have introduced methods that modify the loss function in various ways to address this. However, despite this progress undersampling remains a competitive alternative to these importance weighted classifiers.

Our theory draws from the rich literature on non-parametric classification (Tsybakov, 2010). Apart from borrowing this setting of nonparametric classification, we also utilize upper bounds on the estimation error of the simple histogram estimator (Freedman & Diaconis, 1981; Devroye & Györfi, 1985) to prove our upper bounds in the label shift case. Finally, we note that to prove our minimax lower bounds we proceed by using the general recipe of reducing from estimation to testing (Wainwright, 2019, Chapter 15). One difference from this standard framework is that our training samples shall be drawn from a different distribution than the test samples used to define the risk.

Past work has established lower bounds on the minimax risk for binary classification without distribution shift for general VC classes (see, e.g., Massart & Nédélec, 2006). Note that, these bounds are not directly applicable in the distribution shift setting, and consequently these lower bounds scale with the total number of samples $n = n_{\mathsf{maj}} + n_{\mathsf{min}}$ rather than with the minority number of samples ($n_{\mathsf{min}}$). There are also refinements of this lower bound to obtain minimax lower bounds for cost-sensitive losses that penalize errors on the two class classes differently (Kamalaruban & Williamson, 2018). By carefully selecting these costs it is possible to apply these results in the label shift setting. However, these lower bounds remain loose and decay with $n$ and $n_{\mathsf{maj}}$ in contrast to the tighter $n_{\mathsf{min}}$ dependence in our lower bounds. We provide a more detailed discussion about potentially applying these lower bounds to the label shift setting after the presentation of our theorem in Section 4.1.

There is rich literature that studies domain adaptation and transfer learning under label shift (Maity et al., 2020) and covariate shift (Ben-David et al., 2006; David et al., 2010; Ben-David et al., 2010; Ben-David & Urner, 2012; 2014; Berlind & Urner, 2015; Kpotufe & Martinet, 2018; Hanneke & Kpotufe, 2019). The principal focus of this line of work was to understand the value of unlabeled data from the target domain, rather than to characterize the relative value of the number of labeled samples from the majority and minority groups. Among these papers, most closely related to our work are those in the covariate shift setting (Kpotufe & Martinet, 2018; Hanneke & Kpotufe, 2019). Their lower bound results can be reinterpreted to show that under covariate shift in the absence of overlap, the minimax excess risk is lower bounded by $1/n_{\mathsf{min}}^{1/3}$. We provide a more detailed comparison with their results after presenting our lower bounds in Section 4.2.

Finally, we note that Arjovsky et al. (2022) recently showed that undersampling can improve the worst-class accuracy of linear SVMs in the presence of label shift. In comparison, our results hold for arbitrary classifiers with the rich nonparametric data distributions.

## 3 Setting

In this section, we shall introduce our problem setup and define the types of distribution shift that we consider.

### 3.1 Problem setup

The setting for our study is nonparametric binary classification with Lipschitz data distributions. We are given $n$ training datapoints $\mathcal{S} := \{(x_1, y_1), \ldots, (x_n, y_n)\} \in ([0, 1] \times \{-1, 1\})^n$ that are all drawn from a *train* distribution $\mathsf{P}_{\mathsf{train}}$. During test time, the data shall be drawn from a *different* distribution $\mathsf{P}_{\mathsf{test}}$. Our paper focuses on the robustness to this shift in the distribution from train to test time. To present a clean analysis, we study the case where the features $x$ are bounded scalars, however, it is easy to extend our results to the high-dimensional setting.

Given a classifier $f : [0, 1] \to \{-1, 1\}$, we shall be interested in the test error (risk) of this classifier under the test distribution $\mathsf{P}_{\mathsf{test}}$:

$$R(f; \mathsf{P}_{\mathsf{test}}) := \mathbb{E}_{(x,y) \sim \mathsf{P}_{\mathsf{test}}} \left[ \mathbf{1}(f(x) \neq y) \right].$$

### 3.2 Types of distribution shift

We assume that $\mathsf{P}_{\text{train}}$ consists of a mixture of two groups of unequal size, and $\mathsf{P}_{\text{test}}$ contains equal numbers of samples from both groups. Given a majority group distribution $\mathsf{P}_{\text{maj}}$ and a minority group distribution $\mathsf{P}_{\text{min}}$, the learner has access to $n_{\text{maj}}$ majority group samples and $n_{\text{min}}$ minority group samples:

$$\mathcal{S}_{\text{maj}} \sim \mathsf{P}_{\text{maj}}^{n_{\text{maj}}} \quad \text{and} \quad \mathcal{S}_{\text{min}} \sim \mathsf{P}_{\text{min}}^{n_{\text{min}}}.$$

Here $n_{\text{maj}} > n/2$ and $n_{\text{min}} < n/2$ with $n_{\text{maj}} + n_{\text{min}} = n$. The full training dataset is $\mathcal{S} = \mathcal{S}_{\text{maj}} \cup \mathcal{S}_{\text{min}} = \{(x_1, y_1), \ldots, (x_n, y_n)\}$. We assume that the learner has access to the knowledge whether a particular sample $(x_i, y_i)$ comes from the majority or minority group.

The test samples will be drawn from $\mathsf{P}_{\text{test}} = \frac{1}{2}\mathsf{P}_{\text{maj}} + \frac{1}{2}\mathsf{P}_{\text{min}}$, a uniform mixture over $\mathsf{P}_{\text{maj}}$ and $\mathsf{P}_{\text{min}}$. Thus, the training dataset is an imbalanced draw from the distributions $\mathsf{P}_{\text{maj}}$ and $\mathsf{P}_{\text{min}}$, whereas the test samples are balanced draws. We let $\rho := n_{\text{maj}}/n_{\text{min}} > 1$ denote the imbalance ratio in the training data. We consider the uniform mixture during test time since the resulting test loss is of the same order as the worst-group loss.

We focus on two-types of distribution shifts: label shift and group-covariate shift that we describe below.

#### 3.2.1 Label shift

In this setting, the imbalance in the training data comes from there being more samples from one class over another. Without loss of generality, we shall assume that the class $y = 1$ is the majority class. Then, we define the majority and the minority class distributions as

$$\mathsf{P}_{\text{maj}}(x, y) = \mathsf{P}_1(x)\mathbf{1}(y = 1) \quad \text{and} \quad \mathsf{P}_{\text{min}} = \mathsf{P}_{-1}(x)\mathbf{1}(y = -1),$$

where $\mathsf{P}_1, \mathsf{P}_{-1}$ are class-conditional distributions over the interval $[0, 1]$. We assume that class-conditional distributions $\mathsf{P}_i$ have densities on $[0, 1]$ and that they are 1-Lipschitz: for any $x, x' \in [0, 1]$,

$$|\mathsf{P}_i(x) - \mathsf{P}_i(x')| \le |x - x'|.$$

We denote the class of pairs of distributions $(\mathsf{P}_{\text{maj}}, \mathsf{P}_{\text{min}})$ that satisfy these conditions by $\mathcal{P}_{\text{LS}}$. We note that such Lipschitzness assumptions are common in the literature (see Tsybakov, 2010).

#### 3.2.2 Group-covariate shift

In this setting, we have two groups $\{a, b\}$, and corresponding to each of these groups is a distribution (with densities) over the features $\mathsf{P}_a(x)$ and $\mathsf{P}_b(x)$. We let $a$ correspond to the majority group and $b$ correspond to the minority group. Then, we define

$$\mathsf{P}_{\text{maj}}(x, y) = \mathsf{P}_a(x)\mathsf{P}(y \mid x) \quad \text{and} \quad \mathsf{P}_{\text{min}}(x, y) = \mathsf{P}_b(x)\mathsf{P}(y \mid x).$$

We assume that for $y \in \{-1, 1\}$, for all $x, x' \in [0, 1]$:

$$\left|\mathsf{P}(y \mid x) - \mathsf{P}(y \mid x')\right| \le |x - x'|,$$

that is, the distribution of the label given the feature is 1-Lipschitz, and it varies slowly over the domain.

To quantify the shift between the train and test distribution, we define a notion of overlap between the group distributions $\mathsf{P}_a$ and $\mathsf{P}_b$ as follows:

$$\mathsf{Overlap}(\mathsf{P}_a, \mathsf{P}_b) := 1 - \text{TV}(\mathsf{P}_a, \mathsf{P}_b)$$

where $\text{TV}(\mathsf{P}_a, \mathsf{P}_b) := \sup_{E \subseteq [0,1]} |\mathsf{P}_a(E) - \mathsf{P}_b(E)|$, denotes the total variation distance between $\mathsf{P}_a$ and $\mathsf{P}_b$. Notice that when $\mathsf{P}_a$ and $\mathsf{P}_b$ have disjoint supports, $\text{TV}(\mathsf{P}_a, \mathsf{P}_b) = 1$ and therefore $\mathsf{Overlap}(\mathsf{P}_a, \mathsf{P}_b) = 0$. On the other hand when $\mathsf{P}_a = \mathsf{P}_b$, $\text{TV}(\mathsf{P}_a, \mathsf{P}_b) = 0$ and $\mathsf{Overlap}(\mathsf{P}_a, \mathsf{P}_b) = 1$. When the overlap is 1, the majority and minority distributions are identical and hence we have no shift between train and test. Observe that $\mathsf{Overlap}(\mathsf{P}_a, \mathsf{P}_b) = \mathsf{Overlap}(\mathsf{P}_{\text{maj}}, \mathsf{P}_{\text{min}})$ since $\mathsf{P}(y \mid x)$ is shared across $\mathsf{P}_{\text{maj}}$ and $\mathsf{P}_{\text{min}}$.

Given a level of overlap $\tau \in [0, 1]$ we denote the class of pairs of distributions $(\mathsf{P}_{\mathsf{maj}}, \mathsf{P}_{\mathsf{min}})$ with overlap at least $\tau$ by $\mathcal{P}_{\mathsf{GS}}(\tau)$. It is easy to check that, $\mathcal{P}_{\mathsf{GS}}(\tau) \subseteq \mathcal{P}_{\mathsf{GS}}(0)$ at any overlap level $\tau \in [0, 1]$.

Considering a notion of overlap between the marginal distributions $\mathsf{P}_a(x)$ and $\mathsf{P}_b(x)$ is natural in the group covariate setting since the conditional distribution that we wish to estimate $\mathsf{P}(y \mid x)$ remains constant from train to test time. Higher overlap between $\mathsf{P}_a$ and $\mathsf{P}_b$ allows a classifier to learn more about the underlying conditional distribution $\mathsf{P}(y \mid x)$ when it sees samples from either group. In contrast, in the label shift setting $\mathsf{P}(x \mid y)$ remains constant from train to test time and higher overlap between $\mathsf{P}(x \mid 1)$ and $\mathsf{P}(x \mid -1)$ does not help to estimate $\mathsf{P}(y \mid x)$.

## 4   Lower bounds on the minimax excess risk

In this section, we shall prove our lower bounds that show that the performance of any algorithm is constrained by the number of minority samples $n_{\mathsf{min}}$. Before we state our lower bounds, we need to introduce the notion of excess risk and minimax excess risk.

**Excess risk and minimax excess risk.**  We measure the performance of an algorithm $\mathcal{A}$ through its excess risk defined in the following way. Given an algorithm $\mathcal{A}$ that takes as input a dataset $\mathcal{S}$ and returns a classifier $\mathcal{A}^{\mathcal{S}}$, and a pair of distributions $(\mathsf{P}_{\mathsf{maj}}, \mathsf{P}_{\mathsf{min}})$ with $\mathsf{P}_{\mathsf{test}} = \frac{1}{2}\mathsf{P}_{\mathsf{maj}} + \frac{1}{2}\mathsf{P}_{\mathsf{min}}$, the *expected excess risk* is given by

$$\mathsf{Excess\ Risk}[\mathcal{A}; (\mathsf{P}_{\mathsf{maj}}, \mathsf{P}_{\mathsf{min}})] := \mathbb{E}_{\mathcal{S} \sim \mathsf{P}_{\mathsf{maj}}^{n_{\mathsf{maj}}} \times \mathsf{P}_{\mathsf{min}}^{n_{\mathsf{min}}}} \left[ R(\mathcal{A}^{\mathcal{S}}; \mathsf{P}_{\mathsf{test}})) - R(f^{\star}(\mathsf{P}_{\mathsf{test}}); \mathsf{P}_{\mathsf{test}}) \right], \tag{1}$$

where $f^{\star}(\mathsf{P}_{\mathsf{test}})$ is the Bayes classifier that minimizes the risk $R(\cdot; \mathsf{P}_{\mathsf{test}})$. The first term corresponds to the expected risk for the algorithm when given $n_{\mathsf{maj}}$ samples from $\mathsf{P}_{\mathsf{maj}}$ and $n_{\mathsf{min}}$ samples from $\mathsf{P}_{\mathsf{min}}$, whereas the second term corresponds to the Bayes error for the problem.

Excess risk does not let us characterize the inherent difficulty of a problem, since for any particular data distribution $(\mathsf{P}_{\mathsf{maj}}, \mathsf{P}_{\mathsf{min}})$ the best possible algorithm $\mathcal{A}$ to minimize the excess risk would be the trivial mapping $\mathcal{A}^{\mathcal{S}} = f^{\star}(\mathsf{P}_{\mathsf{test}})$. Therefore, to prove meaningful lower bounds on the performance of algorithms we need to define the notion of minimax excess risk (see Wainwright, 2019, Chapter 15). Given a class of pairs of distributions $\mathcal{P}$ define

$$\mathsf{Minimax\ Excess\ Risk}(\mathcal{P}) := \inf_{\mathcal{A}} \sup_{(\mathsf{P}_{\mathsf{maj}}, \mathsf{P}_{\mathsf{min}}) \in \mathcal{P}} \mathsf{Excess\ Risk}[\mathcal{A}; (\mathsf{P}_{\mathsf{maj}}, \mathsf{P}_{\mathsf{min}})], \tag{2}$$

where the infimum is over all measurable estimators $\mathcal{A}$. The minimax excess risk is the excess risk of the "best" algorithm in the worst case over the class of problems defined by $\mathcal{P}$.

### 4.1   Label shift lower bounds

We demonstrate the hardness of the label shift problem in general by establishing a lower bound on the minimax excess risk.

**Theorem 4.1.** *Consider the label shift setting described in Section 3.2.1. Recall that $\mathcal{P}_{\mathsf{LS}}$ is the class of pairs of distributions $(\mathsf{P}_{\mathsf{maj}}, \mathsf{P}_{\mathsf{min}})$ that satisfy the assumptions in that section. The minimax excess risk over this class is lower bounded as follows:*

$$\mathsf{Minimax\ Excess\ Risk}(\mathcal{P}_{\mathsf{LS}}) = \inf_{\mathcal{A}} \sup_{(\mathsf{P}_{\mathsf{maj}}, \mathsf{P}_{\mathsf{min}}) \in \mathcal{P}_{\mathsf{LS}}} \mathsf{Excess\ Risk}[\mathcal{A}; (\mathsf{P}_{\mathsf{maj}}, \mathsf{P}_{\mathsf{min}})] \geq \frac{1}{600} \frac{1}{n_{\mathsf{min}}^{1/3}}. \tag{3}$$

We establish this result in Appendix B. We show that rather surprisingly, the lower bound on the minimax excess risk scales only with the number of minority class samples $n_{\mathsf{min}}^{1/3}$, and does not depend on $n_{\mathsf{maj}}$. Intuitively, this is because any learner must predict which class-conditional distribution $(\mathsf{P}(x \mid 1)$ or $\mathsf{P}(x \mid -1))$ assigns higher likelihood at that $x$. To interpret this result, consider the extreme scenario where $n_{\mathsf{maj}} \to \infty$ but $n_{\mathsf{min}}$ is finite. In this case, the learner has full information about the majority class distribution. However,

the learning task continues to be challenging since any learner would be uncertain about whether the minority class distribution assigns higher or lower likelihood at any given $x$. This uncertainty underlies the reason why the minimax rate of classification is constrained by the number of minority samples $n_{\mathsf{min}}$.

We briefly note that, applying minimax lower bounds from the transfer learning literature (Maity et al., 2020, Theorem 3.1 with $\alpha = 1$, $\beta = 0$ and $d = 1$) to our problem leads to a more optimistic lower bound of $1/n^{1/3}$. Our lower bounds that scale as $1/n_{\mathsf{min}}^{1/3}$, uncover the fact that only adding minority class samples helps reduce the risk.

As noted above in the introduction, it is possible to obtain lower bounds for the label shift setting by applying bounds from the cost-sensitive classification literature. However, as we shall argue below they are loose and predict the incorrect trend when applied in this setting. Consider the result (Kamalaruban & Williamson, 2018, Theorem 4) which is a minimax lower bound for cost sensitive binary classification that applies to VC classses (which does not capture the nonparameteric setting studied here but it is illuminating to study how that bound scales with the imbalance ratio $\rho = n_{\mathsf{maj}}/n_{\mathsf{min}}$). Assume that the joint distribution during training is a mixture distribution given by $\mathsf{P} = \frac{\rho}{1+\rho}\mathsf{P}_{\mathsf{maj}} + \frac{1}{1+\rho}\mathsf{P}_{\mathsf{min}}$ so that on average the ratio of the number of samples from the majority and minority class is equal to $\rho$. Then by applying their lower bound we find that it scales with $1/(n\rho)$ (see Appendix E for a detailed calculation). This scales inversely with $\rho$ the imbalance ratio and incorrectly predicts that the problem gets easier as the imbalance is larger. In contrast, our lower bound scales with $1/n_{\mathsf{min}} = (1+\rho)/n$, which correctly predicts that as the imbalance is larger, the minimax test error is higher.

## 4.2 Group-covariate shift lower bounds

Next, we shall state our lower bound on the minimax excess risk that demonstrates the hardness of the group-covariate shift problem.

**Theorem 4.2.** *Consider the group shift setting described in Section 3.2.2. Given any overlap $\tau \in [0, 1]$ recall that $\mathcal{P}_{\mathsf{GS}}(\tau)$ is the class of distributions such that $\mathsf{Overlap}(\mathsf{P}_{\mathsf{maj}}, \mathsf{P}_{\mathsf{min}}) \geq \tau$. The minimax excess risk in this setting is lower bounded as follows:*

$$\mathsf{Minimax\ Excess\ Risk}(\mathcal{P}_{\mathsf{GS}}(\tau)) = \inf_{\mathcal{A}} \sup_{(\mathsf{P}_{\mathsf{maj}}, \mathsf{P}_{\mathsf{min}}) \in \mathcal{P}_{\mathsf{GS}}(\tau)} \mathsf{Excess\ Risk}[\mathcal{A}; (\mathsf{P}_{\mathsf{maj}}, \mathsf{P}_{\mathsf{min}})]$$

$$\geq \frac{1}{200(n_{\mathsf{min}} \cdot (2 - \tau) + n_{\mathsf{maj}} \cdot \tau)^{1/3}} \geq \frac{1}{200 n_{\mathsf{min}}^{1/3}(\rho \cdot \tau + 2)^{1/3}}, \tag{4}$$

*where $\rho = n_{\mathsf{maj}}/n_{\mathsf{min}} > 1$.*

We prove this theorem in Appendix C.

We see that in the *low overlap* setting ($\tau \ll 1/\rho$), the minimax excess risk is lower bounded by $1/n_{\mathsf{min}}^{1/3}$, and we are fundamentally constrained by the number of samples in minority group. To see why this is the case, consider the extreme example with $\tau = 0$ where $\mathsf{P}_a$ has support $[0, 0.5]$ and $\mathsf{P}_b$ has support $[0.5, 1]$. The $n_{\mathsf{maj}}$ majority group samples from $\mathsf{P}_a$ provide information about the correct label predict in the interval $[0, 0.5]$ (the support of $\mathsf{P}_a$). However, since the distribution $\mathsf{P}(y \mid x)$ is 1-Lipschitz in the worst case these samples provide very limited information about the correct predictions in $[0.5, 1]$ (the support of $\mathsf{P}_b$). Thus, predicting on the support of $\mathsf{P}_b$ requires samples from the minority group and this results in the $n_{\mathsf{min}}$ dependent rate. In fact, in this extreme case ($\tau = 0$) even if $n_{\mathsf{maj}} \to \infty$, the minimax excess risk is still bounded away from zero. This intuition also carries over to the case when the overlap is small but non-zero and our lower bound shows that minority samples are much more valuable than majority samples at reducing the risk.

On the other hand, when the overlap is high ($\tau \gg 1/\rho$) the minimax excess risk is lower bounded by $1/(n_{\mathsf{min}}(2 - \tau) + n_{\mathsf{maj}}\tau)^{1/3}$ and the extra majority samples are quite beneficial. This is roughly because the supports of $\mathsf{P}_a$ and $\mathsf{P}_b$ have large overlap and hence samples from the majority group are useful in helping make predictions even in regions where $\mathsf{P}_b$ is large. In the extreme case when $\tau = 1$, we have that $\mathsf{P}_a = \mathsf{P}_b$ and therefore recover the classic i.i.d. setting with no distribution shift. Here, the lower bound scales with $1/n^{1/3}$, as one might expect.

Previous work on transfer learning with covariate shift has considered other more elaborate notions of *transferability* (Kpotufe & Martinet, 2018; Hanneke & Kpotufe, 2019) than overlap between group distributions considered here. In the case of no overlap ($\tau = 0$), previous results (Kpotufe & Martinet, 2018, Theorem 1 with $\alpha = 1, \beta = 0$ and $\gamma = \infty$) yield the same lower bound of $1/n_{\min}^{1/3}$. On the other extreme, applying their result (Kpotufe & Martinet, 2018, Theorem 1 with $\alpha = 1, \beta = 0$ and $\gamma = 0$) in the high transfer regime yields a lower bound on $1/n^{1/3}$. This result is aligned with the high overlap $\tau = 1$ case that we consider here.

Beyond these two edge cases of no overlap ($\tau = 0$) and high overlap ($\tau = 1$), our lower bound is key to drawing the simple complementary conclusion that even when overlap between group distributions is small as compared to $1/\rho$, minority samples alone dictate the rate of convergence.

## 5  Upper bounds on the excess risk for the undersampled binning estimator

We will show that an undersampled estimator matches the rates in the previous section showing that undersampling is an optimal robustness intervention. We start by defining the undersampling procedure and the undersampling binning estimator.

**Undersampling procedure.**  Given training data $\mathcal{S} := \{(x_1, y_1), \dots, (x_n, y_n)\}$, generate a new undersampled dataset $\mathcal{S}_{\mathsf{US}}$ by

- including all $n_{\min}$ samples from $\mathcal{S}_{\min}$ and,

- including $n_{\min}$ samples from $\mathcal{S}_{\mathsf{maj}}$ by sampling uniformly at random without replacement.

This procedure ensures that in the undersampled dataset $\mathcal{S}_{\mathsf{US}}$, the groups are balanced, and that $|\mathcal{S}_{\mathsf{US}}| = 2n_{\min}$.

The undersampling binning estimator defined next will first run this undersampling procedure to obtain $\mathcal{S}_{\mathsf{US}}$ and just uses these samples to output a classifier.

**Undersampled binning estimator**  The undersampled binning estimator $\mathcal{A}_{\mathsf{USB}}$ takes as input a dataset $\mathcal{S}$ and a positive integer $K$ corresponding to the number of bins, and returns a classifier $\mathcal{A}_{\mathsf{USB}}^{\mathcal{S},K} : [0, 1] \to \{-1, 1\}$. This estimator is defined as follows:

1. First, we compute the undersampled dataset $\mathcal{S}_{\mathsf{US}}$.

2. Given this dataset $\mathcal{S}_{\mathsf{US}}$, let $n_{1,j}$ be the number of points with label $+1$ that lie in the interval $I_j = [\frac{j-1}{K}, \frac{j}{K}]$. Also, define $n_{-1,j}$ analogously. Then set

$$
\mathcal{A}_j = \begin{cases} 1 & \text{if } n_{1,j} > n_{-1,j}, \\ -1 & \text{otherwise.} \end{cases}
$$

3. Define the classifier $\mathcal{A}_{\mathsf{USB}}^{\mathcal{S},K}$ such that if $x \in I_j$ then

$$
\mathcal{A}_{\mathsf{USB}}^{\mathcal{S},K}(x) = \mathcal{A}_j. \tag{5}
$$

   Essentially in each bin $I_j$, we set the prediction to be the majority label among the samples that fall in this bin.

Whenever the number of bins $K$ is clear from the context we shall denote $\mathcal{A}_{\mathsf{USB}}^{\mathcal{S},K}$ by $\mathcal{A}_{\mathsf{USB}}^{\mathcal{S}}$. Below we establish upper bounds on the excess risk of this simple estimator.

## 5.1 Label shift upper bounds

We now establish an upper bound on the excess risk of $\mathcal{A}_{\mathsf{USB}}$ in the label shift setting (see Section 3.2.1). Below we let $c, C > 0$ be absolute constants independent of problem parameters like $n_{\mathsf{maj}}$ and $n_{\mathsf{min}}$.

**Theorem 5.1.** *Consider the label shift setting described in Section 3.2.1. For any* $(\mathsf{P}_{\mathsf{maj}}, \mathsf{P}_{\mathsf{min}}) \in \mathcal{P}_{\mathsf{LS}}$ *the expected excess risk of the Undersampling Binning Estimator (Eq.* (5)*) with number of bins with* $K = c\lceil n_{\mathsf{min}}^{1/3} \rceil$ *is upper bounded by*

$$\mathsf{Excess\ Risk}[\mathcal{A}_{\mathsf{USB}}; (\mathsf{P}_{\mathsf{maj}}, \mathsf{P}_{\mathsf{min}})] = \mathbb{E}_{\mathcal{S} \sim \mathsf{P}_{\mathsf{maj}}^{n_{\mathsf{maj}}} \times \mathsf{P}_{\mathsf{min}}^{n_{\mathsf{min}}}} \left[ R(\mathcal{A}_{\mathsf{USB}}^{\mathcal{S}}; \mathsf{P}_{\mathsf{test}}) - R(f^\star; \mathsf{P}_{\mathsf{test}}) \right] \leq \frac{C}{n_{\mathsf{min}}^{1/3}}.$$

We prove this result in Appendix B. This upper bound combined with the lower bound in Theorem 4.1 shows that an undersampling approach is minimax optimal up to constants in the presence of label shift.

Our analysis leaves open the possibility of better algorithms when the learner has additional information about the structure of the label shift beyond Lipschitz continuity.

## 5.2 Group-covariate shift upper bounds

Next, we present our upper bounds on the excess risk of the undersampled binning estimator in the group-covariate shift setting (see Section 3.2.2). In the theorem below, $C > 0$ is an absolute constant independent of the problem parameters $n_{\mathsf{maj}}$, $n_{\mathsf{min}}$ and $\tau$.

**Theorem 5.2.** *Consider the group shift setting described in Section 3.2.2. For any overlap* $\tau \in [0, 1]$ *and for any* $(\mathsf{P}_{\mathsf{maj}}, \mathsf{P}_{\mathsf{min}}) \in \mathcal{P}_{\mathsf{GS}}(\tau)$ *the expected excess risk of the Undersampling Binning Estimator (Eq.* (5)*) with number of bins with* $K = \lceil n_{\mathsf{min}}^{1/3} \rceil$ *is*

$$\mathsf{Excess\ Risk}[\mathcal{A}_{\mathsf{USB}}; (\mathsf{P}_{\mathsf{maj}}, \mathsf{P}_{\mathsf{min}})] = \mathbb{E}_{\mathcal{S} \sim \mathsf{P}_{\mathsf{maj}}^{n_{\mathsf{maj}}} \times \mathsf{P}_{\mathsf{min}}^{n_{\mathsf{min}}}} \left[ R(\mathcal{A}_{\mathsf{USB}}^{\mathcal{S}}; \mathsf{P}_{\mathsf{test}})) - R(f^\star; \mathsf{P}_{\mathsf{test}}) \right] \leq \frac{C}{n_{\mathsf{min}}^{1/3}}.$$

We provide a proof for this theorem in Appendix C. Compared to the lower bound established in Theorem 4.2 which scales as $1/\left((2-\tau)n_{\mathsf{min}} + n_{\mathsf{maj}}\tau\right)^{1/3}$, the upper bound for the undersampled binning estimator always scales with $1/n_{\mathsf{min}}^{1/3}$ since it operates on the undersampled dataset ($\mathcal{S}_{\mathsf{US}}$).

Thus, we have shown that in the absence of overlap ($\tau \ll 1/\rho = n_{\mathsf{min}}/n_{\mathsf{maj}}$) there is an undersampling algorithm that is minimax optimal up to constants. However when there is high overlap ($\tau \gg 1/\rho$) there is a non-trivial gap between the upper and lower bounds:

$$\frac{\mathsf{Upper\ Bound}}{\mathsf{Lower\ Bound}} = c(\rho \cdot \tau + 2)^{1/3}.$$

# 6 Minority sample dependence in practice

Inspired by our worst-case theoretical predictions in nonparametric classification, we ask: how does the accuracy of neural network classifiers trained using robust algorithms evolve as a function of the majority and minority samples?

To explore this question, we conduct a small case study using the imbalanced binary CIFAR10 dataset (Byrd & Lipton, 2019; Wang et al., 2022) that is constructed using the "cat" and "dog" classes. The test set consists of all of the 1000 cat and 1000 dog test examples. To form our initial train and validation sets, we take 2500 cat examples but only 500 dog examples from the official train set, corresponding to a 5:1 label imbalance. We then use 80% of those examples for training and the rest for validation. In our experiment, we either (*a*) add only minority samples; (*b*) add only majority samples; (*c*) add both majority and minority samples in a 5:1 ratio. We consider competitive robust classifiers proposed in the literature that are convolutional neural networks trained either by using (*i*) the importance weighted cross entropy loss, or (*ii*) the importance weighted VS loss (Kini et al., 2021). We early stop using the importance weighted validation loss in both cases. The additional experimental details are presented in Appendix G.

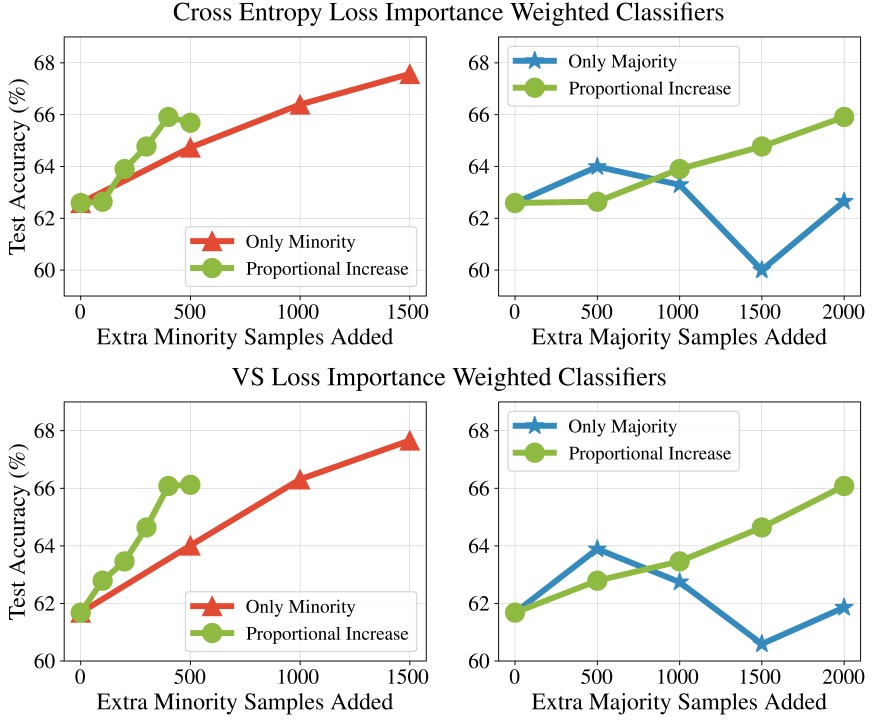

Figure 2: Convolutional neural network classifiers trained on the Imbalanced Binary CIFAR10 dataset with a 5:1 label imbalance. (Top) Models trained using the importance weighted cross entropy loss with early stopping. (Bottom) Models trained using the importance weighted VS loss (Kini et al., 2021) with early stopping. We report the average test accuracy calculated on a balanced test set over 5 random seeds. We start off with 2500 cat examples and 500 dog examples in the training dataset. We find that in accordance with our theory, for both of the classifiers adding only minority class samples (red) leads to large gain in accuracy ($\sim 6\%$), while adding majority class samples (blue) leads to little or no gain. In fact, adding majority samples sometimes hurts test accuracy due to the added bias. When we add majority and minority samples in a 5:1 ratio (green), the gain is largely due to the addition of minority samples and is only marginally higher ($< 2\%$) than adding only minority samples. The green curves correspond to the same classifiers in both the left and right panels.

Our results in Figure 2 are generally consistent with our theoretical predictions. By adding only minority class samples the test accuracy of both classifiers increases by a great extent (6%), while by adding only majority class samples the test accuracy remains constant or in some cases even decreases owing to the added bias of the classifiers. When we add samples to both groups proportionately, the increase in the test accuracy appears to largely to be due to the increase in the number of minority class samples. We see this on the left panels, where the difference between adding only extra minority group samples (red) and both minority and majority group samples (green) is small. Thus, we find that the accuracy for these neural network classifiers is also constrained by the number of minority class samples. Similar conclusions hold for classifiers trained using the tilted loss (Li et al., 2020) and group-DRO objective (Sagawa et al., 2020) (see Appendix D).

# 7  Discussion

We showed that undersampling is an optimal robustness intervention in nonparametric classification in the absence of significant overlap between group distributions or without additional structure beyond Lipschitz continuity. We worked in one dimension for the sake of clarity and it would be interesting to extend this study to higher dimensions. We focused on Lipschitz continuous distributions here, but it is also interesting to consider other forms of regularity such as Hölder continuity.

At a high level our results highlight the need to reason about the specific structure in the distribution shift and design algorithms that are tailored to take advantage of this structure. This would require us to step away from the common practice in robust machine learning where the focus is to design "universal" robustness interventions that are agnostic to the structure in the shift. Alongside this, our results also dictate the need for datasets and benchmarks with the propensity for transfer from train to test time.

**Acknowledgments**

We would like to thank Ke Alexander Wang for his useful comments and feedback in the early stages of this project. We would also like to thank Shibani Santurkar and Dimitrios Tsipras for useful discussions and encouragement. Finally, we would like to thank the anonymous reviewers whose many helpful comments improved the paper. NC was supported by a SAIL Postdoctoral Fellowship and TH was supported by a gift from Open Philanthropy.

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

# A  Technical tools

In this section we avail ourselves of some technical tools that shall be used in all of the proofs below.

## A.1  Reduction to lower bounds over a finite class

The lower bound on the minimax excess risk will be established via the usual route of first identifying a "hard" finite set of problem instances and then establishing the lower bound over this finite class. One difference from the usual setup in proving such lower bounds (see Wainwright, 2019, Chapter 15) is that the training samples are drawn from an imbalanced distribution, whereas the test samples are drawn from a balanced one.

Let $\mathcal{P}$ be a class of pairs of distributions, where each element $(\mathsf{P}_{\mathsf{maj}}, \mathsf{P}_{\mathsf{min}}) \in \mathcal{P}$ is a pair of distributions over $[0,1] \times \{-1,1\}$. As before, we let $\mathsf{P}_{\mathsf{test}}$ denote the uniform mixture over $\mathsf{P}_{\mathsf{maj}}$ and $\mathsf{P}_{\mathsf{min}}$. We let $\mathcal{V}$ denote a finite index set. Corresponding to each element $v \in \mathcal{V}$ there is a $\mathsf{P}_v = (\mathsf{P}_{v,\mathsf{maj}}, \mathsf{P}_{v,\mathsf{min}}) \in \mathcal{P}$ with $\mathsf{P}_{v,\mathsf{test}} = (\mathsf{P}_{v,\mathsf{maj}} + \mathsf{P}_{v,\mathsf{min}})/2$. Finally, also define a pair of random variables $(V, S)$ as follows:

1. $V$ is a uniform random variable over the set $\mathcal{V}$.

2. $(S \mid V = v) \sim \mathsf{P}_{v,\mathsf{maj}}^{n_{\mathsf{maj}}} \times \mathsf{P}_{v,\mathsf{min}}^{n_{\mathsf{min}}}$, is an independent draw of $n_{\mathsf{maj}}$ samples from $\mathsf{P}_{v,\mathsf{maj}}$ and $n_{\mathsf{min}}$ samples from $\mathsf{P}_{v,\mathsf{min}}$.

We shall let $\mathsf{Q}$ denote the joint distribution of the random variables $(V, S)$, and let $\mathsf{Q}_S$ denote the marginal distribution of $S$.

With this notation in place, we now present a lemma that lower bounds the minimax excess risk in terms of quantities defined over the finite class of "hard" instances $\mathsf{P}_v$.

**Lemma A.1.** *Let the random variables $(V, S)$ be as defined above. The minimax excess risk is lower bounded as follows:*

$$\text{Minimax Excess Risk}(\mathcal{P}) = \inf_{\mathcal{A}} \sup_{(\mathsf{P}_{\text{maj}}, \mathsf{P}_{\text{min}}) \in \mathcal{P}} \mathbb{E}_{\mathcal{S} \sim \mathsf{P}_{\text{maj}}^{n_{\text{maj}}} \times \mathsf{P}_{\text{min}}^{n_{\text{min}}}} \left[ R(\mathcal{A}^{\mathcal{S}}; \mathsf{P}_{\text{test}}) - R(f^{\star}(\mathsf{P}_{\text{test}}); \mathsf{P}_{\text{test}}) \right]$$
$$\geq \mathfrak{R}_{\mathcal{V}} - \mathfrak{B}_{\mathcal{V}},$$

*where $\mathfrak{R}_{\mathcal{V}}$ and Bayes-error $\mathfrak{B}_{\mathcal{V}}$ are defined as*

$$\mathfrak{R}_{\mathcal{V}} := \mathbb{E}_{S \sim \mathsf{Q}_S} [\inf_h \Pr_{(x,y) \sim \sum_{v \in \mathcal{V}} \mathsf{Q}(v|S) \mathsf{P}_{v, \text{test}}} (h(x) \neq y)],$$
$$\mathfrak{B}_{\mathcal{V}} := \mathbb{E}_V [R(f^{\star}(\mathsf{P}_{V, \text{test}}); \mathsf{P}_{V, \text{test}}))].$$

*Proof.* By the definition of Minimax Excess Risk,

$$\text{Minimax Excess Risk} = \inf_{\mathcal{A}} \sup_{(\mathsf{P}_{\text{maj}}, \mathsf{P}_{\text{min}}) \in \mathcal{P}} \mathbb{E}_{\mathcal{S} \sim \mathsf{P}_{\text{maj}}^{n_{\text{maj}}} \times \mathsf{P}_{\text{min}}^{n_{\text{min}}}} [R(\mathcal{A}^{\mathcal{S}}; \mathsf{P}_{\text{test}})] - R(f^{\star}(\mathsf{P}_{\text{test}}); \mathsf{P}_{\text{test}})$$
$$\geq \inf_{\mathcal{A}} \sup_{v \in \mathcal{V}} \mathbb{E}_{S|v \sim \mathsf{P}_{v,\text{maj}}^{n_{\text{maj}}} \times \mathsf{P}_{v,\text{min}}^{n_{\text{min}}}} [R(\mathcal{A}^S; \mathsf{P}_{v, \text{test}})] - R(f^{\star}(\mathsf{P}_{v, \text{test}}); \mathsf{P}_{v, \text{test}})$$
$$\geq \inf_{\mathcal{A}} \mathbb{E}_V \left[ \mathbb{E}_{S|V \sim \mathsf{P}_{V,\text{maj}}^{n_{\text{maj}}} \times \mathsf{P}_{V,\text{min}}^{n_{\text{min}}}} [R(\mathcal{A}^S; \mathsf{P}_{V, \text{test}})] - R(f^{\star}(\mathsf{P}_{V, \text{test}}); \mathsf{P}_{V, \text{test}})) \right]$$
$$= \inf_{\mathcal{A}} \mathbb{E}_V [\mathbb{E}_{S|V \sim \mathsf{P}_{V,\text{maj}}^{n_{\text{maj}}} \times \mathsf{P}_{V,\text{min}}^{n_{\text{min}}}} [R(\mathcal{A}^S; \mathsf{P}_{V, \text{test}})]] - \underbrace{\mathbb{E}_V [R(f^{\star}(\mathsf{P}_{V, \text{test}}); \mathsf{P}_{V, \text{test}}))]}_{= \mathfrak{B}_{\mathcal{V}}}.$$

We continue lower bounding the first term as follows

$$\inf_{\mathcal{A}} \mathbb{E}_V [\mathbb{E}_{S|V \sim \mathsf{P}_{V,\text{maj}}^{n_{\text{maj}}} \times \mathsf{P}_{V,\text{min}}^{n_{\text{min}}}} [R(\mathcal{A}^S; \mathsf{P}_{V, \text{test}})]] = \inf_{\mathcal{A}} \mathbb{E}_{(V,S) \sim \mathsf{Q}} [\Pr_{(x,y) \sim \mathsf{P}_{V, \text{test}}} (\mathcal{A}^S(x) \neq y)]$$
$$= \inf_{\mathcal{A}} \mathbb{E}_{S \sim \mathsf{Q}_S} \mathbb{E}_{V \sim \mathsf{Q}(\cdot|S)} [\Pr_{(x,y) \sim \mathsf{P}_{V, \text{test}}} (\mathcal{A}^S(x) \neq y)]$$
$$\overset{(i)}{\geq} \mathbb{E}_{S \sim \mathsf{Q}_S} [\inf_h \mathbb{E}_{V \sim \mathsf{Q}(\cdot|S)} [\Pr_{(x,y) \sim \mathsf{P}_{V, \text{test}}} (h(x) \neq y)]]$$
$$= \mathbb{E}_{S \sim \mathsf{Q}_S} [\inf_h \Pr_{(x,y) \sim \sum_{v \in \mathcal{V}} \mathsf{Q}(v|S) \mathsf{P}_{v, \text{test}}} (h(x) \neq y)]$$
$$= \mathfrak{R}_{\mathcal{V}},$$

where $(i)$ follows since $\mathcal{A}^S$ is a fixed classifier given the sample set $S$. This, combined with the previous equation block completes the proof. $\square$

## A.2 The hat function and its properties

In this section, we define the *hat function* and establish some of its properties. This function will be useful in defining "hard" problem instances to prove our lower bounds. Given a positive integer $K$ the hat function is defined as

$$\phi_K(x) = \begin{cases} |x + \frac{1}{4K}| - \frac{1}{4K} & \text{for } x \in [-\frac{1}{2K}, 0], \\ \frac{1}{4K} - |x - \frac{1}{4K}| & \text{for } x \in [0, \frac{1}{2K}], \\ 0 & \text{otherwise}. \end{cases} \quad (6)$$

When $K$ is clear from context, we omit the subscript.

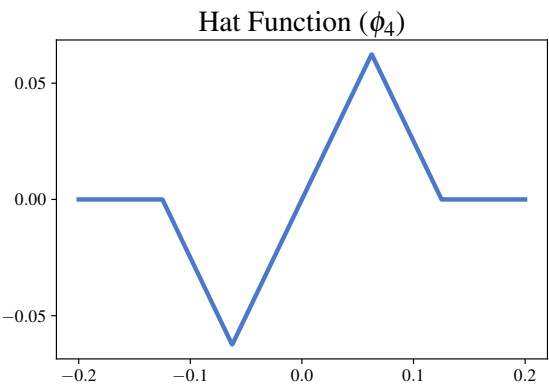

Figure 3: The hat function with $K = 4$.

We first notice that this function is 1-Lipschitz and odd, so

$$\int_{-\frac{1}{2K}}^{\frac{1}{2K}} \phi_K(x) \, \mathrm{d}x = 0.$$

We also compute some other key quantities for $\phi$.

**Lemma A.2.** *For any positive integer $K$,*

$$\int_{-\frac{1}{2K}}^{\frac{1}{2K}} |\phi_K(x)| \, \mathrm{d}x = \frac{1}{8K^2}.$$

*Proof.* We suppress $K$ in the notation. We have that,

$$\int_{-\frac{1}{2K}}^{\frac{1}{2K}} |\phi(x)| \, \mathrm{d}x = \int_{-\frac{1}{2K}}^{0} \left| \frac{1}{4K} - \left| x + \frac{1}{4K} \right| \right| \, \mathrm{d}x + \int_{0}^{\frac{1}{2K}} \left| \left| x - \frac{1}{4K} \right| - \frac{1}{4K} \right| \, \mathrm{d}x.$$

The integrand $\left| \frac{1}{4K} - \left| x + \frac{1}{4K} \right| \right|$ over $x \in \left[ -\frac{1}{2K}, 0 \right]$ defines a triangle with base $\frac{1}{2K}$ and height $\frac{1}{4K}$, thus it has area $\frac{1}{16K^2}$. Therefore,

$$\int_{-\frac{1}{2K}}^{0} \left| \frac{1}{4K} - \left| x + \frac{1}{4K} \right| \right| \, \mathrm{d}x = \frac{1}{16K^2}.$$

The same holds for the second term. Thus, by adding them up we get that $\int_{-\frac{1}{2K}}^{\frac{1}{2K}} |\phi(x)| \, \mathrm{d}x = \frac{1}{8K^2}$. $\qquad \square$

**Lemma A.3.** *For any positive integer $K$,*

$$\int_{0}^{\frac{1}{K}} \log \left( \frac{1 + \phi_K(x - \frac{1}{2K})}{1 - \phi_K(x - \frac{1}{2K})} \right) \left( 1 + \phi_K \left( x - \frac{1}{2K} \right) \right) \, \mathrm{d}x \leq \frac{1}{3K^3}$$

*and*

$$\int_{0}^{\frac{1}{K}} \log \left( \frac{1 - \phi_K(x - \frac{1}{2K})}{1 + \phi_K(x - \frac{1}{2K})} \right) \left( 1 - \phi_K \left( x - \frac{1}{2K} \right) \right) \, \mathrm{d}x \leq \frac{1}{3K^3}.$$

*Proof.* Let us suppress $K$ in the notation. We prove the first bound below and the second bound follows by an identical argument. We have that

$$
\int_0^{\frac{1}{K}} \log\left(\frac{1 + \phi(x - \frac{1}{2K})}{1 - \phi(x - \frac{1}{2K})}\right) \left(1 + \phi\left(x - \frac{1}{2K}\right)\right) \, \mathrm{d}x
$$

$$
= \int_{-\frac{1}{2K}}^{\frac{1}{2K}} \log\left(\frac{1 + \phi(x)}{1 - \phi(x)}\right) (1 + \phi(x)) \, \mathrm{d}x
$$

$$
= \int_0^{\frac{1}{2K}} \log\left(\frac{1 + \phi(x)}{1 - \phi(x)}\right) (1 + \phi(x)) \, \mathrm{d}x + \int_{-\frac{1}{2K}}^0 \log\left(\frac{1 + \phi(x)}{1 - \phi(x)}\right) (1 + \phi(x)) \, \mathrm{d}x
$$

$$
= \int_0^{\frac{1}{2K}} \log\left(\frac{1 + \phi(x)}{1 - \phi(x)}\right) (1 + \phi(x)) \, \mathrm{d}x - \int_{\frac{1}{2K}}^0 \log\left(\frac{1 + \phi(-x)}{1 - \phi(-x)}\right) (1 + \phi(-x)) \, \mathrm{d}x
$$

$$
= \int_0^{\frac{1}{2K}} \log\left(\frac{1 + \phi(x)}{1 - \phi(x)}\right) (1 + \phi(x)) \, \mathrm{d}x + \int_0^{\frac{1}{2K}} \log\left(\frac{1 - \phi(x)}{1 + \phi(x)}\right) (1 - \phi(x)) \, \mathrm{d}x,
$$

where the last equality follows since $\phi$ is an odd function. Now, we may collect the integrands to get that,

$$
\int_0^{\frac{1}{K}} \log\left(\frac{1 + \phi(x - \frac{1}{2K})}{1 - \phi(x - \frac{1}{2K})}\right) \left(1 + \phi\left(x - \frac{1}{2K}\right)\right) \, \mathrm{d}x
$$

$$
= 2\int_0^{\frac{1}{2K}} \log\left(\frac{1 + \phi(x)}{1 - \phi(x)}\right) \phi(x) \, \mathrm{d}x
$$

$$
= 2\int_0^{\frac{1}{2K}} \log\left(1 + \frac{2\phi(x)}{1 - \phi(x)}\right) \phi(x) \, \mathrm{d}x
$$

$$
\leq 2\int_0^{\frac{1}{2K}} \frac{2\phi(x)^2}{1 - \phi(x)} \, \mathrm{d}x,
$$

where the last inequality follows since $\log(1 + x) \leq x$ for all $x$. Now we observe that $\phi(x) \leq x \leq \frac{1}{2}$ for $x \in [0, \frac{1}{2K}]$, and in particular, $\frac{1}{1 - \phi(x)} \leq 2$. Thus,

$$
\int_0^{\frac{1}{K}} \log\left(\frac{1 + \phi(x - \frac{1}{2K})}{1 - \phi(x - \frac{1}{2K})}\right) \left(1 + \phi\left(x - \frac{1}{2K}\right)\right) \, \mathrm{d}x
$$

$$
\leq 8\int_0^{\frac{1}{2K}} \phi(x)^2 \, \mathrm{d}x
$$

$$
\leq 8\int_0^{\frac{1}{2K}} x^2 \, \mathrm{d}x
$$

$$
= \frac{1}{3K^3}.
$$

This proves the first bound. The second bound follows analogously. $\square$

## B Proofs in the label shift setting

Throughout this section we operate in the label shift setting (Section 3.2.1).

First, in Appendix B.1 through a sequence of lemmas we prove the minimax lower bound Theorem 4.1. Next, in Appendix B.2 we prove Theorem 5.1 which is an upper bound on the excess risk of the undersampled binning estimator (see Eq. (5)) with $\lceil n_{\min} \rceil^{1/3}$ bins by invoking previous results on nonparametric density estimation (Freedman & Diaconis, 1981; Devroye & Györfi, 1985).

### B.1 Proof of Theorem 4.1

In this section, we provide a proof of the minimax lower bound in the label shift setting.

We will proceed by constructing a class of distributions where the separation between any two distributions in the class is small enough such that it is hard to distinguish between them with finite minority class samples. In particular, we split the interval $[0, 1]$ into sub-intervals and each class distribution on each sub-interval either has slightly more probability mass on the left side of the sub-interval, on the right, or completely uniform. Since the minority class sample size is limited, no classifier will be able to tell which distribution the minority class is generated from, and hence will suffer high excess risk.

We construct the "hard" set of distributions as follows. Fix $K$ to be an integer that will be specified in the sequel as a function of $n_{\mathsf{min}}$. Let the index set be $\mathcal{V} = \{-1, 0, 1\}^K \times \{-1, 0, 1\}^K$. For $v \in \mathcal{V}$, we will let $v_1 \in \{-1, 0, 1\}^K$ be the first $K$ coordinates and $v_{-1} \in \{-1, 0, 1\}^K$ be the last $K$ coordinates. That is, $v = (v_1, v_{-1})$.

For every $v \in \mathcal{P}$ we shall define pair of class-conditional distributions $\mathsf{P}_{v,1}$ and $\mathsf{P}_{v,-1}$ as follows: for $x \in I_j = [\frac{j-1}{K}, \frac{j}{K}]$,

$$\mathsf{P}_{v,1}(x) = 1 + v_{1,j}\phi\left(x - \frac{j+1/2}{K}\right)$$
$$\mathsf{P}_{v,-1}(x) = 1 + v_{-1,j}\phi\left(x - \frac{j+1/2}{K}\right),$$

where $\phi$ is defined in Eq. 6. Notice that $\mathsf{P}_{v,1}$ only depends on $v_1$ while $\mathsf{P}_{v,-1}$ only depends on $v_{-1}$. We continue to define

$$\mathsf{P}_{v,\mathsf{maj}}(x, y) = \mathsf{P}_{v,1}(x)\mathbf{1}(y = 1)$$
$$\mathsf{P}_{v,\mathsf{min}}(x, y) = \mathsf{P}_{v,-1}(x)\mathbf{1}(y = -1),$$

and

$$\mathsf{P}_{v,\mathsf{test}}(x, y) = \frac{\mathsf{P}_{v,\mathsf{maj}}(x, y) + \mathsf{P}_{v,\mathsf{min}}(x, y)}{2} = \frac{\mathsf{P}_{v,1}(x)\mathbf{1}(y = 1) + \mathsf{P}_{v,-1}(x)\mathbf{1}(y = -1)}{2}.$$

Observe that in the test distribution it is equally likely for the label to be $+1$ or $-1$.

Recall that as described in Section A.1, $V$ shall be a uniform random variable over $\mathcal{V}$ and $S \mid V \sim \mathsf{P}_{v,\mathsf{maj}}^{n_{\mathsf{maj}}} \times \mathsf{P}_{v,\mathsf{min}}^{n_{\mathsf{min}}}$. We shall let $\mathsf{Q}$ denote the joint distribution of $(V, S)$ and let $\mathsf{Q}_S$ denote the marginal over $S$.

With this construction in place, we first show that the minimax excess risk is lower bounded as follows.

**Lemma B.1.** *For any positive integers $K, n_{\mathsf{maj}}, n_{\mathsf{min}}$, the minimax excess risk is lower bounded as follows:*

$$\mathsf{Minimax\ Excess\ Risk}(\mathcal{P}_{\mathsf{LS}})$$
$$= \inf_{\mathcal{A}} \sup_{(\mathsf{P}_{\mathsf{maj}}, \mathsf{P}_{\mathsf{min}}) \in \mathcal{P}_{\mathsf{LS}}} \mathbb{E}_{S \sim \mathsf{P}_{\mathsf{maj}}^{n_{\mathsf{maj}}} \times \mathsf{P}_{\mathsf{min}}^{n_{\mathsf{min}}}} \left[ R(\mathcal{A}^S; \mathsf{P}_{\mathsf{test}}) - R(f^\star; \mathsf{P}_{\mathsf{test}}) \right]$$
$$\geq \frac{1}{36K} - \frac{1}{2}\mathbb{E}_{S \sim \mathsf{Q}_S}\left[ \mathsf{TV}\left( \sum_{v \in \mathcal{V}} \mathsf{Q}(v \mid S)\mathsf{P}_{v,1}, \sum_{v \in \mathcal{V}} \mathsf{Q}(v \mid S)\mathsf{P}_{v,-1} \right) \right]. \tag{7}$$

*Proof.* By invoking Lemma A.1 we get that

$$\mathsf{Minimax\ Excess\ Risk}(\mathcal{P}_{\mathsf{LS}})$$
$$\geq \mathbb{E}_{S \sim \mathsf{Q}_S}\underbrace{[\inf_h \Pr_{(x,y) \sim \sum_{v \in \mathcal{V}} \mathsf{Q}(v|S)\mathsf{P}_{v,\mathsf{test}}}(h(x) \neq y)]}_{=:\mathfrak{R}_{\mathcal{V}}} - \underbrace{\mathbb{E}_V[R(f^\star(\mathsf{P}_{V,\mathsf{test}}); \mathsf{P}_{V,\mathsf{test}}))]}_{=:\mathfrak{B}_{\mathcal{V}}}.$$

We proceed by calculating alternate expressions for $\mathfrak{R}_{\mathcal{V}}$ and $\mathfrak{B}_{\mathcal{V}}$ to get our desired lower bound on the minimax excess risk.

**Calculation of $\mathfrak{R}_{\mathcal{V}}$:** Immediately by Le Cam's lemma (Wainwright, 2019, Eq. 15.13), we get that

$$\mathfrak{R}_{\mathcal{V}} = \mathbb{E}_{S \sim \mathsf{Q}_S} \left[ \inf_{h} \Pr_{(x,y) \sim \sum_{v \in \mathcal{V}} \mathsf{Q}(v|S)\mathsf{P}_{v,\text{test}}} (h(x) \neq y) \right]$$

$$= \frac{1}{2} \mathbb{E}_{S \sim \mathsf{Q}_S} \left[ 1 - \text{TV} \left( \sum_{v \in \mathcal{V}} \mathsf{Q}(v \mid S)\mathsf{P}_{v,1}, \sum_{v \in \mathcal{V}} \mathsf{Q}(v \mid S)\mathsf{P}_{v,-1} \right) \right]. \tag{8}$$

**Calculation of $\mathfrak{B}_{\mathcal{V}}$:** Again by invoking Le Cam's lemma (Wainwright, 2019, Eq. 15.13), we get that for any class conditional distributions $\mathsf{P}_1, \mathsf{P}_{-1}$,

$$R(f^{\star}; \mathsf{P}_{\text{test}}) = \frac{1}{2} - \frac{1}{2}\text{TV}(\mathsf{P}_1, \mathsf{P}_{-1}).$$

So by taking expectations, we get that

$$\mathfrak{B}_{\mathcal{V}} = \mathbb{E}_V[R(f^{\star}(\mathsf{P}_{V,\text{test}}); \mathsf{P}_{V,\text{test}})] = \mathbb{E}_V \left[ \frac{1}{2} - \frac{1}{2}\text{TV}(\mathsf{P}_{V,1}, \mathsf{P}_{V,-1}) \right]. \tag{9}$$

We now compute $\mathbb{E}_V[\text{TV}(\mathsf{P}_{V,1}, \mathsf{P}_{V,-1})]$ as follows:

$$\mathbb{E}_V[\text{TV}(\mathsf{P}_{V,1}, \mathsf{P}_{V,-1})] = \frac{1}{2}\mathbb{E}_V \left[ \int_{x=0}^{1} |\mathsf{P}_{V,1}(x) - \mathsf{P}_{V,-1}(x)| \, dx \right]$$

$$= \frac{1}{2}\mathbb{E}_V \left[ \sum_{j=1}^{K} \int_{\frac{j-1}{K}}^{\frac{j}{K}} |V_{1,j} - V_{-1,j}| \left| \phi\left(x - \frac{j+1/2}{K}\right) \right| \, dx \right]$$

$$= \frac{1}{2}\sum_{j=1}^{K} \mathbb{E}_V \left[ \int_{\frac{j-1}{K}}^{\frac{j}{K}} |V_{1,j} - V_{-1,j}| \left| \phi\left(x - \frac{j+1/2}{K}\right) \right| \, dx \right]$$

$$\overset{(i)}{=} \frac{1}{16K^2}\sum_{j=1}^{K} \mathbb{E}_V[|V_{1,j} - V_{-1,j}|],$$

where $(i)$ follows by Lemma A.2. Observe that $V_{1,j}, V_{-1,j}$ are independent uniform random variables on $\{-1, 0, 1\}$, it is therefore straightforward to compute that

$$\mathbb{E}_V[|V_{1,j} - V_{-1,j}|] = \frac{8}{9}.$$

This yields that

$$\mathbb{E}_V[\text{TV}(\mathsf{P}_{V,1}, \mathsf{P}_{V,-1})] = \frac{1}{18K}.$$

Plugging this into Eq. (9) allows us to conclude that

$$\mathfrak{B}_{\mathcal{V}} = \mathbb{E}_V[R(f^{\star}(\mathsf{P}_{V,\text{test}}); \mathsf{P}_{V,\text{test}})] = \frac{1}{2}\left(1 - \frac{1}{18K}\right). \tag{10}$$

Combining Eqs. (8) and (10) establishes the claimed result.

$\square$

In light of this previous lemma we now aim to upper bound the expected total variation distance in Eq. (7).

**Lemma B.2.** *Suppose that $v$ is drawn uniformly from the set $\{-1, 1\}^K$, and that $S \mid v$ is drawn from $\mathsf{P}_{v,\text{maj}}^{n_{\text{maj}}} \times \mathsf{P}_{v,\text{min}}^{n_{\text{min}}}$ then,*

$$\mathbb{E}_S \left[ \text{TV} \left( \sum_{v \in \mathcal{V}} \mathsf{Q}(v \mid S)\mathsf{P}_{v,1}, \sum_{v \in \mathcal{V}} \mathsf{Q}(v \mid S)\mathsf{P}_{v,-1} \right) \right] \leq \frac{1}{18K} - \frac{1}{144K}\exp\left(-\frac{n_{\text{min}}}{3K^3}\right).$$

*Proof.* Let $\psi := \mathbb{E}_S \left[ \text{TV} \left( \sum_{v \in \mathcal{V}} \mathsf{Q}(v \mid S) \mathsf{P}_{v,1}, \sum_{v \in \mathcal{V}} \mathsf{Q}(v \mid S) \mathsf{P}_{v,-1} \right) \right]$. Then,

$$
\begin{aligned}
\psi &= \mathbb{E}_S \left[ \text{TV} \left( \sum_{v \in \mathcal{V}} \mathsf{Q}(v \mid S) \mathsf{P}_{v,1}, \sum_{v \in \mathcal{V}} \mathsf{Q}(v \mid S) \mathsf{P}_{v,-1} \right) \right] \\
&= \frac{1}{2} \mathbb{E}_S \left[ \int_{x=0}^{1} \left| \sum_{v \in \mathcal{V}} \mathsf{Q}(v \mid S) \left( \mathsf{P}_{v,1}(x) - \mathsf{P}_{v,-1}(x) \right) \right| \, dx \right] \\
&= \frac{1}{2} \mathbb{E}_S \left[ \sum_{j=1}^{K} \int_{x=\frac{j-1}{K}}^{\frac{j}{K}} \left| \sum_{v \in \mathcal{V}} \mathsf{Q}(v \mid S) \left( \mathsf{P}_{v,1}(x) - \mathsf{P}_{v,-1}(x) \right) \right| \, dx \right] \\
&= \frac{1}{2} \mathbb{E}_S \left[ \sum_{j=1}^{K} \int_{x=\frac{j-1}{K}}^{\frac{j}{K}} \left| \sum_{v \in \mathcal{V}} \mathsf{Q}(v \mid S)(v_{1,j} - v_{-1,j}) \phi \left( x - \frac{j+1/2}{K} \right) \right| \, dx \right],
\end{aligned}
$$

where the last equality is by the definition of $\mathsf{P}_{v,1}$ and $\mathsf{P}_{v,-1}$. Continuing we get that,

$$
\begin{aligned}
\psi &= \frac{1}{2} \sum_{j=1}^{K} \left[ \int_{x=\frac{j-1}{K}}^{\frac{j}{K}} \left| \phi \left( x - \frac{j+1/2}{K} \right) \right| \, dx \right] \mathbb{E}_S \left[ \left| \sum_{v \in \mathcal{V}} \mathsf{Q}(v \mid S)(v_{1,j} - v_{-1,j}) \right| \right] \\
&\overset{(i)}{=} \frac{1}{16K^2} \mathbb{E}_S \left[ \sum_{j=1}^{K} \left| \sum_{v \in \mathcal{V}} \mathsf{Q}(v \mid S)(v_{1,j} - v_{-1,j}) \right| \right] \\
&= \frac{1}{16K^2} \sum_{j=1}^{K} \int \left| \sum_{v \in \mathcal{V}} \mathsf{Q}(v \mid S)(v_{1,j} - v_{-1,j}) \right| \, d\mathsf{Q}_S(S) \\
&= \frac{1}{16K^2} \sum_{j=1}^{K} \int \left| \sum_{v \in \mathcal{V}} \mathsf{Q}(v, S)(v_{1,j} - v_{-1,j}) \right| \, dS \\
&\overset{(ii)}{=} \frac{1}{16K^2 |\mathcal{V}|} \sum_{j=1}^{K} \int \left| \sum_{v \in \mathcal{V}} \mathsf{Q}(S \mid v)(v_{1,j} - v_{-1,j}) \right| \, dS,
\end{aligned}
$$

where $(i)$ follows by the calculation in Lemma A.2 and $(ii)$ follows since $v$ is a uniform random variable over the set $\mathcal{V}$.

The distributions $\mathsf{P}_{v,1}$ and $\mathsf{P}_{v,-1}$ are symmetrically defined over all intervals $I_j = [\frac{j-1}{K}, \frac{j}{K}]$, and hence all of the summands in the RHS above are equal. Thus,

$$
\psi = \frac{1}{16K |\mathcal{V}|} \int \left| \sum_{v \in \mathcal{V}} \mathsf{Q}(S \mid v)(v_{1,1} - v_{-1,1}) \right| \, dS. \tag{11}
$$

Before we continue further, let us define

$$
\mathcal{V}^+ = \{v \in \mathcal{V} \mid v_{1,1} > v_{-1,1}\}.
$$

For every $v \in \mathcal{V}^+$, let $\tilde{v} \in \mathcal{V}$ be such that is the same as $v$ on all coordinates, except $\tilde{v}_{1,1} = -v_{1,1}$ and $\tilde{v}_{-1,1} = -v_{-1,1}$. Then continuing from Eq. (11) we find that,

$$
\begin{aligned}
\psi &\stackrel{(i)}{=} \frac{1}{16K|\mathcal{V}|} \int \left| \sum_{v \in \mathcal{V}^+} (v_{1,1} - v_{-1,1})(\mathsf{Q}(S \mid v) - \mathsf{Q}(S \mid \tilde{v})) \right| \mathrm{d}S \\
&\stackrel{(ii)}{\leq} \frac{1}{16K|\mathcal{V}|} \int \sum_{v \in \mathcal{V}^+} (v_{1,1} - v_{-1,1}) \left| \mathsf{Q}(S \mid v) - \mathsf{Q}(S \mid \tilde{v}) \right| \mathrm{d}S \\
&= \frac{1}{16K|\mathcal{V}|} \sum_{v \in \mathcal{V}^+} (v_{1,1} - v_{-1,1}) \int \left| \mathsf{Q}(S \mid v) - \mathsf{Q}(S \mid \tilde{v}) \right| \mathrm{d}S \\
&= \frac{1}{8K|\mathcal{V}|} \underbrace{\sum_{v \in \mathcal{V}^+} (v_{1,1} - v_{-1,1}) \mathrm{TV}(\mathsf{Q}(S \mid v), \mathsf{Q}(S \mid \tilde{v}))}_{=: \Xi},
\end{aligned}
$$
(12)

where $(i)$ we use the definition of $\mathcal{V}^+$ and $\tilde{v}$, $(ii)$ follows since $v_{1,1} > v_{-1,1}$ for $v \in \mathcal{V}^+$.

Now we further partition $\mathcal{V}^+$ into 3 sets $\mathcal{V}^{(1,0)}, \mathcal{V}^{(0,-1)}, \mathcal{V}^{(1,-1)}$ as follows

$$
\begin{aligned}
\mathcal{V}^{(1,0)} &= \{v \in \mathcal{V} \mid v_{1,1} = 1, v_{-1,1} = 0\}, \\
\mathcal{V}^{(0,-1)} &= \{v \in \mathcal{V} \mid v_{1,1} = 0, v_{-1,1} = -1\}, \\
\mathcal{V}^{(1,-1)} &= \{v \in \mathcal{V} \mid v_{1,1} = 1, v_{-1,1} = -1\}.
\end{aligned}
$$

Note that $\mathsf{Q}(S \mid v) = \mathsf{P}_{v,\mathsf{maj}}^{n_{\mathsf{maj}}} \times \mathsf{P}_{v,\mathsf{min}}^{n_{\mathsf{min}}}$, and therefore

$$
\begin{aligned}
\Xi &= \sum_{v \in \mathcal{V}^+} (v_{1,1} - v_{-1,1}) \mathrm{TV}\left( \mathsf{P}_{v,\mathsf{maj}}^{n_{\mathsf{maj}}} \times \mathsf{P}_{v,\mathsf{min}}^{n_{\mathsf{min}}}, \mathsf{P}_{\tilde{v},\mathsf{maj}}^{n_{\mathsf{maj}}} \times \mathsf{P}_{\tilde{v},\mathsf{min}}^{n_{\mathsf{min}}} \right) \\
&\stackrel{(i)}{=} \sum_{v \in \mathcal{V}^{(1,0)}} \mathrm{TV}\left( \mathsf{P}_{v,\mathsf{maj}}^{n_{\mathsf{maj}}} \times \mathsf{P}_{v,\mathsf{min}}^{n_{\mathsf{min}}}, \mathsf{P}_{\tilde{v},\mathsf{maj}}^{n_{\mathsf{maj}}} \times \mathsf{P}_{\tilde{v},\mathsf{min}}^{n_{\mathsf{min}}} \right) \\
&\quad + \sum_{v \in \mathcal{V}^{(0,-1)}} \mathrm{TV}\left( \mathsf{P}_{v,\mathsf{maj}}^{n_{\mathsf{maj}}} \times \mathsf{P}_{v,\mathsf{min}}^{n_{\mathsf{min}}}, \mathsf{P}_{\tilde{v},\mathsf{maj}}^{n_{\mathsf{maj}}} \times \mathsf{P}_{\tilde{v},\mathsf{min}}^{n_{\mathsf{min}}} \right) \\
&\quad + 2 \sum_{v \in \mathcal{V}^{(1,-1)}} \mathrm{TV}\left( \mathsf{P}_{v,\mathsf{maj}}^{n_{\mathsf{maj}}} \times \mathsf{P}_{v,\mathsf{min}}^{n_{\mathsf{min}}}, \mathsf{P}_{\tilde{v},\mathsf{maj}}^{n_{\mathsf{maj}}} \times \mathsf{P}_{\tilde{v},\mathsf{min}}^{n_{\mathsf{min}}} \right),
\end{aligned}
$$
(13)

where $(i)$ follows since $v_1, v_{-1} \in \{-1, 0, 1\}^K$ and by the definition of the sets $\mathcal{V}^{(1,0)}, \mathcal{V}^{(0,1)}$ and $\mathcal{V}^{(1,-1)}$.

Now by the Bretagnolle–Huber inequality (see Canonne, 2022, Corollary 4),

$$
\begin{aligned}
\mathrm{TV}\left( \mathsf{P}_{v,\mathsf{maj}}^{n_{\mathsf{maj}}} \times \mathsf{P}_{v,\mathsf{min}}^{n_{\mathsf{min}}}, \mathsf{P}_{\tilde{v},\mathsf{maj}}^{n_{\mathsf{maj}}} \times \mathsf{P}_{\tilde{v},\mathsf{min}}^{n_{\mathsf{min}}} \right) &= \mathrm{TV}\left( \mathsf{P}_{\tilde{v},\mathsf{maj}}^{n_{\mathsf{maj}}} \times \mathsf{P}_{\tilde{v},\mathsf{min}}^{n_{\mathsf{min}}}, \mathsf{P}_{v,\mathsf{maj}}^{n_{\mathsf{maj}}} \times \mathsf{P}_{v,\mathsf{min}}^{n_{\mathsf{min}}} \right) \\
&\leq 1 - \frac{1}{2} \exp\left( -\mathrm{KL}\left( \mathsf{P}_{\tilde{v},\mathsf{maj}}^{n_{\mathsf{maj}}} \times \mathsf{P}_{\tilde{v},\mathsf{min}}^{n_{\mathsf{min}}} \| \mathsf{P}_{v,\mathsf{maj}}^{n_{\mathsf{maj}}} \times \mathsf{P}_{v,\mathsf{min}}^{n_{\mathsf{min}}} \right) \right),
\end{aligned}
$$

where we flip the arguments in the first step for simplicity later.

Next, by the chain rule for KL-divergence, we have that

$$
\mathrm{KL}(\mathsf{P}_{\tilde{v},\mathsf{maj}}^{n_{\mathsf{maj}}} \times \mathsf{P}_{\tilde{v},\mathsf{min}}^{n_{\mathsf{min}}} \| \mathsf{P}_{v,\mathsf{maj}}^{n_{\mathsf{maj}}} \times \mathsf{P}_{v,\mathsf{min}}^{n_{\mathsf{min}}}) = n_{\mathsf{maj}} \mathrm{KL}(\mathsf{P}_{\tilde{v},\mathsf{maj}} \| \mathsf{P}_{v,\mathsf{maj}}) + n_{\mathsf{min}} \mathrm{KL}(\mathsf{P}_{\tilde{v},\mathsf{min}} \| \mathsf{P}_{v,\mathsf{min}}).
$$

Using these, let us upper bound the first term in Eq. (13) corresponding to $v \in \mathcal{V}^{(0,-1)}$. For $v \in \mathcal{V}^{(0,-1)}$, notice that $\mathrm{KL}(\mathsf{P}_{\tilde{v},\mathsf{maj}} \| \mathsf{P}_{v,\mathsf{maj}}) = 0$ since $v_{1,j} = \tilde{v}_{1,j}$ for all $j \in \{1, \ldots, K\}$. For the second term, $\mathrm{KL}(\mathsf{P}_{\tilde{v},\mathsf{min}} \| \mathsf{P}_{v,\mathsf{min}})$,

only $v_{1,1}$ and $\tilde{v}_{1,1}$ differ, so

$$
\begin{aligned}
\mathrm{KL}(\mathsf{P}_{\tilde{v},\mathrm{min}}\|\mathsf{P}_{v,\mathrm{min}}) &= \int_0^1 \mathsf{P}_{v,-1}(x) \log\left(\frac{\mathsf{P}_{v,-1}(x)}{\mathsf{P}_{\tilde{v},-1}(x)}\right) \, \mathrm{d}x \\
&= \int_0^{\frac{1}{K}} \log\left(\frac{1 + \phi_K(x - \frac{1}{2K})}{1 - \phi_K(x - \frac{1}{2K})}\right) \left(1 + \phi_K\left(x - \frac{1}{2K}\right)\right) \, \mathrm{d}x \\
&\leq \frac{1}{3K^3},
\end{aligned}
$$

where the last inequality is a result of the calculation in Lemma A.3.

Therefore, we get

$$
\sum_{v \in \mathcal{V}^{(0,-1)}} \mathrm{TV}\left(\mathsf{P}_{v,\mathrm{maj}}^{n_{\mathrm{maj}}} \times \mathsf{P}_{v,\mathrm{min}}^{n_{\mathrm{min}}}, \mathsf{P}_{\tilde{v},\mathrm{maj}}^{n_{\mathrm{maj}}} \times \mathsf{P}_{\tilde{v},\mathrm{min}}^{n_{\mathrm{min}}}\right) \leq 9^{K-1}\left(1 - \frac{1}{2}\exp\left(-\frac{n_{\mathrm{min}}}{3K^3}\right)\right).
$$

For the terms in Eq. (13) corresponding to $\mathcal{V}^{(0,-1)}, \mathcal{V}^{(1,-1)}$, we simply take the trivial bound to get

$$
\sum_{v \in \mathcal{V}^{(0,-1)}} \mathrm{TV}\left(\mathsf{P}_{v,\mathrm{maj}}^{n_{\mathrm{maj}}} \times \mathsf{P}_{v,\mathrm{min}}^{n_{\mathrm{min}}}, \mathsf{P}_{\tilde{v},\mathrm{maj}}^{n_{\mathrm{maj}}} \times \mathsf{P}_{\tilde{v},\mathrm{min}}^{n_{\mathrm{min}}}\right) \leq 9^{K-1},
$$

$$
\sum_{v \in \mathcal{V}^{(1,-1)}} \mathrm{TV}\left(\mathsf{P}_{v,\mathrm{maj}}^{n_{\mathrm{maj}}} \times \mathsf{P}_{v,\mathrm{min}}^{n_{\mathrm{min}}}, \mathsf{P}_{\tilde{v},\mathrm{maj}}^{n_{\mathrm{maj}}} \times \mathsf{P}_{\tilde{v},\mathrm{min}}^{n_{\mathrm{min}}}\right) \leq 9^{K-1}.
$$

Plugging these bounds into Eq. (13) we get that,

$$
\Xi \leq 4 \cdot 9^{K-1} - \frac{9^{K-1}}{2}\exp\left(-\frac{n_{\mathrm{min}}}{3K^3}\right).
$$

Now using this bound on $\Xi$ in Eq. (12) and observing that $|\mathcal{V}| = 9^K$, we get that,

$$
\begin{aligned}
\psi &= \mathbb{E}_S\left[\mathrm{TV}\left(\sum_{v \in \mathcal{V}} Q(v \mid S)P_{v,1}, \sum_{v \in \mathcal{V}} Q(v \mid S)P_{v,-1}\right)\right] \\
&\leq \frac{1}{8 \cdot 9^K K}\left(4 \cdot 9^{K-1} - \frac{9^{K-1}}{2}\exp\left(-\frac{n_{\mathrm{min}}}{3K^3}\right)\right) \\
&= \frac{1}{18K} - \frac{1}{144K}\exp\left(-\frac{n_{\mathrm{min}}}{3K^3}\right),
\end{aligned}
$$

completing the proof. $\qquad\square$

Finally, we combine Lemma B.1 and Lemma B.2 to establish the minimax lower bound in this label shift setting. We recall the statement of the theorem here.

**Theorem 4.1.** *Consider the label shift setting described in Section 3.2.1. Recall that $\mathcal{P}_{\mathrm{LS}}$ is the class of pairs of distributions $(\mathsf{P}_{\mathrm{maj}}, \mathsf{P}_{\mathrm{min}})$ that satisfy the assumptions in that section. The minimax excess risk over this class is lower bounded as follows:*

$$
\mathsf{Minimax\ Excess\ Risk}(\mathcal{P}_{\mathrm{LS}}) = \inf_{\mathcal{A}} \sup_{(\mathsf{P}_{\mathrm{maj}},\mathsf{P}_{\mathrm{min}}) \in \mathcal{P}_{\mathrm{LS}}} \mathsf{Excess\ Risk}[\mathcal{A}; (\mathsf{P}_{\mathrm{maj}}, \mathsf{P}_{\mathrm{min}})] \geq \frac{1}{600}\frac{1}{n_{\mathrm{min}}^{1/3}}. \tag{3}
$$

*Proof.* By Lemma B.1 we know that,

$$
\mathsf{Minimax\ Excess\ Risk}(\mathcal{P}_{\mathrm{LS}}) \geq \frac{1}{36K} - \frac{1}{2}\mathbb{E}_{S \sim \mathsf{Q}_S}\left[\mathrm{TV}\left(\sum_{v \in \mathcal{V}} \mathsf{Q}(v \mid S)\mathsf{P}_{v,1}, \sum_{v \in \mathcal{V}} \mathsf{Q}(v \mid S)\mathsf{P}_{v,-1}\right)\right].
$$

Next by the calculation in Lemma B.2 we have that

$$\mathsf{Minimax\ Excess\ Risk}(\mathcal{P}_{\mathsf{LS}}) \geq \frac{1}{36K} - \frac{1}{2}\left(\frac{1}{18K} - \frac{1}{144K}\exp\left(-\frac{n_{\min}}{3K^3}\right)\right)$$
$$= \frac{1}{288K}\exp\left(-\frac{n_{\min}}{3K^3}\right).$$

Setting $K = \lceil n_{\min}^{1/3}\rceil$ yields the following

$$\mathsf{Minimax\ Excess\ Risk}(\mathcal{P}_{\mathsf{LS}}) \geq \frac{1}{288\lceil n_{\min}^{1/3}\rceil}\exp\left(-\frac{n_{\min}}{3\lceil n_{\min}^{1/3}\rceil^3}\right)$$
$$\geq \frac{\exp\left(-\frac{n_{\min}}{3\lceil n_{\min}^{1/3}\rceil^3}\right)}{288}\frac{n_{\min}^{1/3}}{\lceil n_{\min}^{1/3}\rceil}\frac{1}{n_{\min}^{1/3}}$$
$$\overset{(i)}{\geq} \frac{0.7\exp\left(-\frac{1}{3}\right)}{288}\frac{1}{n_{\min}^{1/3}}$$
$$\geq \frac{1}{600}\frac{1}{n_{\min}^{1/3}},$$

where $(i)$ follows since $n_{\min}^{1/3}/\lceil n_{\min}^{1/3}\rceil \geq 0.7$ for $n_{\min} \geq 1$. $\qquad\square$

## B.2 Proof of Theorem 5.1

In this section, we derive an upper bound on the excess risk of the undersampled binning estimator $\mathcal{A}_{\mathsf{USB}}$ (Eq. (5)) in the label shift setting. Recall that given a dataset $\mathcal{S}$ this estimator first calculates the undersampled dataset $\mathcal{S}_{\mathsf{US}}$, where the number of points from the minority group ($n_{\min}$) is equal to the number of points from the majority group ($n_{\min}$), and the size of the dataset is $2n_{\min}$. Throughout this section, $(\mathsf{P}_{\mathsf{maj}}, \mathsf{P}_{\mathsf{min}})$ shall be an arbitrary element of $\mathcal{P}_{\mathsf{LS}}$.

To bound the excess risk of the undersampling algorithm, we will relate it to density estimation.

Recall that $n_{1,j}$ denotes the number of points in $\mathcal{S}_{\mathsf{US}}$ with label $+1$ that lie in $I_j$, and $n_{-1,j}$ is defined analogously.

Given a positive integer $K$, for $x \in I_j = [\frac{j-1}{K}, \frac{j}{K}]$, by the definition of the undersampled binning estimator (Eq. (5))

$$\mathcal{A}_{\mathsf{USB}}^{\mathcal{S}}(x) = \begin{cases} 1 & \text{if } n_{1,j} > n_{-1,j}, \\ -1 & \text{otherwise.} \end{cases}$$

Recall that since we have undersampled, $\sum_j n_{1,j} = \sum_j n_{-1,j} = n_{\min}$. Therefore, define the simple histogram estimators for $\mathsf{P}_1(x) = \mathsf{P}(x \mid y = 1)$ and $\mathsf{P}_{-1}(x) = \mathsf{P}(x \mid y = -1)$ as follows: for $x \in I_j$,

$$\widehat{\mathsf{P}}_1^{\mathcal{S}}(x) := \frac{n_{1,j}}{Kn_{\min}} \quad \text{and} \quad \widehat{\mathsf{P}}_{-1}^{\mathcal{S}}(x) := \frac{n_{-1,j}}{Kn_{\min}}.$$

With this histogram estimator in place, we may define an estimator for $\eta(x) := \mathsf{P}_{\mathsf{test}}(y = 1|x)$ as follows,

$$\widehat{\eta}^{\mathcal{S}}(x) := \frac{\widehat{\mathsf{P}}_1^{\mathcal{S}}(x)}{\widehat{\mathsf{P}}_1^{\mathcal{S}}(x) + \widehat{\mathsf{P}}_{-1}^{\mathcal{S}}(x)}.$$

Observe that, for $x \in I_j$

$$\widehat{\eta}^{\mathcal{S}}(x) > 1/2 \iff n_{1,j} > n_{-1,j} \iff \mathcal{A}_{\mathsf{USB}}^{\mathcal{S}}(x) = 1.$$

Defining an estimator $\widehat{\eta}^{\mathcal{S}}$ for the $\mathsf{P}_{\mathsf{test}}(y = 1 \mid x)$ in this way will allow us to relate the excess risk of $\mathcal{A}_{\mathsf{USB}}$ to the estimation error in $\widehat{\mathsf{P}}_1^{\mathcal{S}}$ and $\widehat{\mathsf{P}}_{-1}^{\mathcal{S}}$.

Before proving the theorem we restate it here.

**Theorem 5.1.** *Consider the label shift setting described in Section 3.2.1. For any $(\mathsf{P}_{\mathsf{maj}}, \mathsf{P}_{\mathsf{min}}) \in \mathcal{P}_{\mathsf{LS}}$ the expected excess risk of the Undersampling Binning Estimator (Eq. (5)) with number of bins with $K = c\lceil n_{\mathsf{min}}^{1/3} \rceil$ is upper bounded by*

$$\text{Excess Risk}[\mathcal{A}_{\mathsf{USB}}; (\mathsf{P}_{\mathsf{maj}}, \mathsf{P}_{\mathsf{min}})] = \mathbb{E}_{\mathcal{S} \sim \mathsf{P}_{\mathsf{maj}}^{n_{\mathsf{maj}}} \times \mathsf{P}_{\mathsf{min}}^{n_{\mathsf{min}}}} \left[ R(\mathcal{A}_{\mathsf{USB}}^{\mathcal{S}}; \mathsf{P}_{\mathsf{test}}) - R(f^\star; \mathsf{P}_{\mathsf{test}}) \right] \leq \frac{C}{n_{\mathsf{min}}^{1/3}}.$$

*Proof.* By the definition of the excess risk

$$\text{Excess Risk}[\mathcal{A}_{\mathsf{USB}}; (\mathsf{P}_{\mathsf{maj}}, \mathsf{P}_{\mathsf{min}})] := \mathbb{E}_{\mathcal{S} \sim \mathsf{P}_{\mathsf{maj}}^{n_{\mathsf{maj}}} \times \mathsf{P}_{\mathsf{min}}^{n_{\mathsf{min}}}} \left[ R(\mathcal{A}_{\mathsf{USB}}^{\mathcal{S}}; \mathsf{P}_{\mathsf{test}})) - R(f^\star; \mathsf{P}_{\mathsf{test}}) \right].$$

By invoking (Wasserman, 2019, Theorem 1) we may upper bound the excess risk given a draw of $\mathcal{S}$ by

$$R(\mathcal{A}_{\mathsf{USB}}^{\mathcal{S}}; \mathsf{P}_{\mathsf{test}})) - R(f^\star; \mathsf{P}_{\mathsf{test}}) \leq 2 \int \left| \widehat{\eta}^{\mathcal{S}}(x) - \eta(x) \right| \mathsf{P}_{\mathsf{test}}(x) \, \mathrm{d}x.$$

Continuing using the definition of $\widehat{\eta}^{\mathcal{S}}$ above and because $\eta = \mathsf{P}_1/(\mathsf{P}_1 + \mathsf{P}_{-1})$ we have that,

$$
\begin{aligned}
& R(\mathcal{A}_{\mathsf{USB}}^{\mathcal{S}}; \mathsf{P}_{\mathsf{test}})) - R(f^\star; \mathsf{P}_{\mathsf{test}}) \\
&= 2 \int_0^1 \left| \frac{\widehat{\mathsf{P}}_1^{\mathcal{S}}(x)}{\widehat{\mathsf{P}}_1^{\mathcal{S}}(x) + \widehat{\mathsf{P}}_{-1}^{\mathcal{S}}(x)} - \frac{\mathsf{P}_1(x)}{\mathsf{P}_1(x) + \mathsf{P}_{-1}(x)} \right| \left( \frac{\mathsf{P}_1(x) + \mathsf{P}_{-1}(x)}{2} \right) \, \mathrm{d}x \\
&= \int_0^1 \left| \left( \frac{\mathsf{P}_1(x) + \mathsf{P}_{-1}(x)}{\widehat{\mathsf{P}}_1^{\mathcal{S}}(x) + \widehat{\mathsf{P}}_{-1}^{\mathcal{S}}(x)} \right) \widehat{\mathsf{P}}_1^{\mathcal{S}}(x) - \mathsf{P}_1(x) \right| \, \mathrm{d}x \\
&\overset{(i)}{\leq} \int_0^1 \left| \widehat{\mathsf{P}}_1^{\mathcal{S}}(x) - \mathsf{P}_1(x) \right| \, \mathrm{d}x + \int_0^1 \left| \frac{\mathsf{P}_1(x) + \mathsf{P}_{-1}(x)}{\widehat{\mathsf{P}}_1^{\mathcal{S}}(x) + \widehat{\mathsf{P}}_{-1}^{\mathcal{S}}(x)} - 1 \right| \widehat{\mathsf{P}}_1^{\mathcal{S}}(x) \, \mathrm{d}x \\
&= \int_0^1 \left| \widehat{\mathsf{P}}_1^{\mathcal{S}}(x) - \mathsf{P}_1(x) \right| \, \mathrm{d}x + \int_0^1 \left| \widehat{\mathsf{P}}_1^{\mathcal{S}}(x) + \widehat{\mathsf{P}}_{-1}^{\mathcal{S}}(x) - \mathsf{P}_1(x) - \mathsf{P}_{-1}(x) \right| \frac{\widehat{\mathsf{P}}_1^{\mathcal{S}}(x)}{\widehat{\mathsf{P}}_1^{\mathcal{S}}(x) + \widehat{\mathsf{P}}_{-1}^{\mathcal{S}}(x)} \, \mathrm{d}x \\
&\leq 2 \int_0^1 \left| \widehat{\mathsf{P}}_1^{\mathcal{S}}(x) - \mathsf{P}_1(x) \right| \, \mathrm{d}x + \int_0^1 \left| \widehat{\mathsf{P}}_{-1}^{\mathcal{S}}(x) - \mathsf{P}_{-1}(x) \right| \, \mathrm{d}x \\
&\overset{(ii)}{\leq} 2 \sqrt{\int_0^1 \left( \widehat{\mathsf{P}}_1^{\mathcal{S}}(x) - \mathsf{P}_1(x) \right)^2 \, \mathrm{d}x} + \sqrt{\int_0^1 \left( \widehat{\mathsf{P}}_{-1}^{\mathcal{S}}(x) - \mathsf{P}_{-1}(x) \right)^2 \, \mathrm{d}x},
\end{aligned}
$$

where $(i)$ follows by the triangle inequality, $(ii)$ is by the Cauchy–Schwarz inequality.

Taking expectation over the samples $\mathcal{S}$ and by invoking Jensen's inequality we find that,

$$
\begin{aligned}
& \text{Excess Risk}(\mathcal{A}^{\mathcal{S}}; (\mathsf{P}_{\mathsf{maj}}, \mathsf{P}_{\mathsf{min}})) \\
&= \mathbb{E}_{\mathcal{S}} \left[ R(\mathcal{A}_{\mathsf{USB}}^{\mathcal{S}}; \mathsf{P}_{\mathsf{test}})) - R(f^\star; \mathsf{P}_{\mathsf{test}}) \right] \\
&\leq 2 \sqrt{\mathbb{E}_{\mathcal{S}} \left[ \int \left( \widehat{\mathsf{P}}_1^{\mathcal{S}}(x) - \mathsf{P}_1(x) \right)^2 \, \mathrm{d}x \right]} + \sqrt{\mathbb{E}_{\mathcal{S}} \left[ \int \left( \widehat{\mathsf{P}}_{-1}^{\mathcal{S}}(x) - \mathsf{P}_{-1}(x) \right)^2 \, \mathrm{d}x \right]}.
\end{aligned}
$$

We note that $\widehat{\mathsf{P}}_j^{\mathcal{S}}$ only depends on $n_{\mathsf{min}}$ i.i.d. draws from class $j$. Thus by (Freedman & Diaconis, 1981, Theorem 1.7), if $K = c\lceil n_{\mathsf{min}} \rceil^{1/3}$ then

$$\mathbb{E}_{\mathcal{S}} \left[ \int \left( \widehat{\mathsf{P}}_j^{\mathcal{S}}(x) - \mathsf{P}_j(x) \right)^2 \, \mathrm{d}x \right] \leq \frac{C}{n_{\mathsf{min}}^{2/3}}.$$

Plugging this into the previous inequality yields the desired result. $\qquad\square$

# C   Proof in the group-covariate shift setting

Throughout this section we operate in the group-covariate shift setting (Section 3.2.2).

We will proceed similarly to Section B. We shall construct a family of class-conditional distributions such that it will be necessary for adequate samples in each sub-interval of $[0, 1]$ to be able to learn the maximally likely label in that sub-interval. On the other hand, we will construct the group-covariate distributions to be separated from one another. As a consequence, sub-intervals with high probability mass under the minority group distribution will have low probability mass under the majority group distribution. Hence, these sub-intervals will not have enough training sample points for any classifier to be able to learn the maximally likely label and as a result shall suffer high excess risk.

First in Appendix C.1, we prove Theorem 4.2, the minimax lower bound through a sequence of lemmas. Second in Appendix C.2, we prove Theorem 5.2 that upper bound on the excess risk of the undersampled binning estimator with $\lceil n_{\min} \rceil^{1/3}$ bins.

## C.1   Proof of Theorem 4.2

In this section, we provide a proof of the minimax lower bound in the group shift setting.

We construct the "hard" set of distributions as follows. Let the index set be $\mathcal{V} = \{-1, 1\}^K$. For every $v \in \mathcal{V}$ define a distribution as follows: for $x \in I_j = [\frac{j-1}{K}, \frac{j}{K}]$,

$$\mathsf{P}_v(y = 1 \mid x) := \frac{1}{2} \left[ 1 + v_j \phi \left( x - \frac{j + 1/2}{K} \right) \right],$$

where $\phi$ is defined in Eq. 6. Given a $\tau \in [0, 1]$ we also construct the group distributions as follows:

$$\mathsf{P}_a(x) = \begin{cases} 2 - \tau & \text{if } x \in [0, 0.5) \\ \tau & \text{if } x \in [0.5, 1], \end{cases}$$

and let

$$\mathsf{P}_b(x) = 2 - \mathsf{P}_a(x).$$

We can verify that

$$\mathsf{Overlap}(\mathsf{P}_a, \mathsf{P}_b) = 1 - \mathrm{TV}(\mathsf{P}_a, \mathsf{P}_b) = 1 - \frac{1}{2} \int_{x=0}^1 |\mathsf{P}_a(x) - \mathsf{P}_b(x)| \, \mathrm{d}x = \tau.$$

We continue to define

$$\mathsf{P}_{v,\mathsf{maj}}(x, y) = \mathsf{P}_v(y \mid x) \mathsf{P}_a(x)$$
$$\mathsf{P}_{v,\mathsf{min}}(x, y) = \mathsf{P}_v(y \mid x) \mathsf{P}_b(x),$$

and

$$\mathsf{P}_{v,\mathsf{test}}(x, y) = \mathsf{P}_v(y \mid x) \left( \frac{\mathsf{P}_a(x) + \mathsf{P}_b(x)}{2} \right).$$

Observe that $(\mathsf{P}_a(x) + \mathsf{P}_b(x))/2 = 1$, the uniform distribution over $[0, 1]$.

Recall that as described in Section A.1, $V$ shall be a uniform random variable over $\mathcal{V}$ and $S \mid V \sim \mathsf{P}_{v,\mathsf{maj}}^{n_{\mathsf{maj}}} \times \mathsf{P}_{v,\mathsf{min}}^{n_{\mathsf{min}}}$. We shall let $\mathsf{Q}$ denote the joint distribution of $(V, S)$ and let $\mathsf{Q}_S$ denote the marginal over $S$.

With this construction in place, we present the following lemma that lower bounds the minimax excess risk by a sum of $\exp(-\mathrm{KL}(\mathsf{Q}(S \mid v_j = 1) \| \mathsf{Q}(S \mid v_j = -1))$ over the intervals. Intuitively, $\mathrm{KL}(\mathsf{Q}(S \mid v_j = 1) \| \mathsf{Q}(S \mid v_j = -1))$ is a measure of how difficult it is to identify whether $v_j = 1$ or $v_j = -1$ from the samples.

**Lemma C.1.** *For any positive integers $K, n_{\mathsf{maj}}, n_{\mathsf{min}}$ and $\tau \in [0, 1]$, the minimax excess risk is lower bounded as follows:*

$$\text{Minimax Excess Risk}(\mathcal{P}_{\mathsf{GS}}(\tau)) = \inf_{\mathcal{A}} \sup_{(\mathsf{P}_{\mathsf{maj}}, \mathsf{P}_{\mathsf{min}}) \in \mathcal{P}_{\mathsf{GS}}(\tau)} \mathbb{E}_{S \sim \mathsf{P}_{\mathsf{maj}}^{n_{\mathsf{maj}}} \times \mathsf{P}_{\mathsf{min}}^{n_{\mathsf{min}}}} \left[ R(\mathcal{A}^S; \mathsf{P}_{\mathsf{test}}) - R(f^\star; \mathsf{P}_{\mathsf{test}}) \right]$$

$$\geq \frac{1}{32K^2} \sum_{j=1}^{K} \exp(-\mathrm{KL}(\mathsf{Q}(S \mid v_j = 1) \| \mathsf{Q}(S \mid v_j = -1))).$$

*Proof.* By invoking Lemma A.1, we know that the minimax excess risk is lower bounded by

$$\text{Minimax Excess Risk}(\mathcal{P}_{\mathsf{GS}}(\tau))$$

$$\geq \underbrace{\mathbb{E}_{S \sim \mathsf{Q}_S} [\inf_{h} \Pr_{(x,y) \sim \sum_{v \in \mathcal{V}} \mathsf{Q}(v|S) \mathsf{P}_{v,\mathsf{test}}} (h(x) \neq y)]}_{=\mathfrak{R}_{\mathcal{V}}} - \underbrace{\mathbb{E}_V [R(f^\star(\mathsf{P}_{V,\mathsf{test}}); \mathsf{P}_{V,\mathsf{test}})]}_{=\mathfrak{B}_{\mathcal{V}}},$$

where $V$ is a uniform random variable over the set $\mathcal{V}$, $S \mid V = v$ is a draw from $\mathsf{P}_{v,\mathsf{maj}}^{n_{\mathsf{maj}}} \times \mathsf{P}_{v,\mathsf{min}}^{n_{\mathsf{min}}}$, and $\mathsf{Q}$ denotes the joint distribution over $(V, S)$.

We shall lower bound this minimax risk in parts. First, we shall establish a lower bound on $\mathfrak{R}_{\mathcal{V}}$, and then an upper bound on the Bayes risk $\mathfrak{B}_{\mathcal{V}}$.

**Lower bound on $\mathfrak{R}_{\mathcal{V}}$.** Unpacking $\mathfrak{R}_{\mathcal{V}}$ using its definition we get that,

$$\mathfrak{R}_{\mathcal{V}} = \mathbb{E}_{S \sim \mathsf{Q}_S} [\inf_{h} \Pr_{(x,y) \sim \sum_{v \in \mathcal{V}} \mathsf{Q}(v|S) \mathsf{P}_{v,\mathsf{test}}} (h(x) \neq y)]$$

$$= \mathbb{E}_{S \sim \mathsf{Q}_S} \left[ \inf_{h} \int_0^1 \mathsf{P}_{\mathsf{test}}(x) \Pr_{y \sim \sum_{v \in \mathcal{V}} \mathsf{Q}(v|S) \mathsf{P}_v(\cdot|x)} [h(x) \neq y] \, \mathrm{d}x \right]$$

$$\overset{(i)}{=} \mathbb{E}_{S \sim \mathsf{Q}_S} \left[ \int_0^1 \mathsf{P}_{\mathsf{test}}(x) \min \left\{ \sum_{v \in \mathcal{V}} \mathsf{Q}(v \mid S) \mathsf{P}_v(1 \mid x), \sum_{v \in \mathcal{V}} \mathsf{Q}(v \mid S) \mathsf{P}_v(-1 \mid x) \right\} \mathrm{d}x \right]$$

$$\overset{(ii)}{=} \frac{1}{2} - \mathbb{E}_{S \sim \mathsf{Q}_S} \left[ \int_0^1 \mathsf{P}_{\mathsf{test}}(x) \left| \frac{1}{2} - \sum_{v \in \mathcal{V}} \mathsf{Q}(v \mid S) \mathsf{P}_v(1 \mid x) \right| \mathrm{d}x \right]$$

$$\overset{(iii)}{=} \frac{1}{2} - \int_0^1 \mathsf{P}_{\mathsf{test}}(x) \mathbb{E}_{S \sim \mathsf{Q}_S} \left[ \left| \frac{1}{2} - \sum_{v \in \mathcal{V}} \mathsf{Q}(v \mid S) \mathsf{P}_v(1 \mid x) \right| \right] \mathrm{d}x, \tag{14}$$

where $(i)$ follows by taking $h$ to be the pointwise minimizer over $x$, $(ii)$ follows since $\mathsf{P}_v(-1 \mid x) = 1 - \mathsf{P}_v(1 \mid x)$ and $\min\{s, 1 - s\} = (1 - |1 - 2s|)/2$ for all $s \in [0, 1]$, and $(iii)$ follows by Fubini's theorem which allows us to switch the order of the integrals.

If $x \in I_j = [\frac{j-1}{K}, \frac{j}{K}]$ for some $j \in \{1, \dots, K\}$ we let $j_x$ denote the value of this index $j$. With this notation in place let us continue to upper bound integrand in the second term in the RHS above as follows:

$$\mathbb{E}_{S \sim \mathsf{Q}_S} \left[ \left| \frac{1}{2} - \sum_{v \in \mathcal{V}} \mathsf{Q}(v \mid S) \mathsf{P}_v(1 \mid x) \right| \right]$$

$$\overset{(i)}{=} \mathbb{E}_{S \sim \mathsf{Q}_S} \left[ \left| \phi \left( x - \frac{j_x + 1/2}{K} \right) \right| |\mathsf{Q}(v_{j_x} = 1 \mid S) - \mathsf{Q}(v_{j_x} = -1 \mid S)| \right]$$

$$= \left| \phi \left( x - \frac{j_x + 1/2}{K} \right) \right| \mathbb{E}_{S \sim \mathsf{Q}_S} [|\mathsf{Q}(v_{j_x} = 1 \mid S) - \mathsf{Q}(v_{j_x} = -1 \mid S)|]$$

$$\overset{(ii)}{=} \left| \phi \left( x - \frac{j_x + 1/2}{K} \right) \right| \mathbb{E}_{S \sim \mathsf{Q}_S} \left[ \left| \frac{\mathsf{Q}(S \mid v_{j_x} = 1) \mathsf{Q}_V(v_{j_x} = 1)}{\mathsf{Q}_S(S)} - \frac{\mathsf{Q}(S \mid v_{j_x} = -1) \mathsf{Q}_V(v_{j_x} = -1)}{\mathsf{Q}_S(S)} \right| \right]$$

$$\overset{(iii)}{=} \frac{1}{2} \left| \phi \left( x - \frac{j_x + 1/2}{K} \right) \right| \mathrm{TV}(\mathsf{Q}(S \mid v_{j_x} = 1), \mathsf{Q}(S \mid v_{j_x} = -1)), \tag{15}$$

where $(i)$ follows since $\mathsf{P}_v(1 \mid x) = (1 + v_{j_x}\phi(x - (j_x + 1/2)/K))/2$ and by marginalizing $\mathsf{Q}(v \mid S)$ over the indices $j \neq j_x$, $(ii)$ follows by using Bayes' rule and $(iii)$ follows since the total-variation distance is half the $\ell_1$ distance. Now by the Bretagnolle–Huber inequality (see Canonne, 2022, Corollary 4) we get that,

$$\mathrm{TV}(\mathsf{Q}(S \mid v_{j_x} = 1), \mathsf{Q}(S \mid v_{j_x} = -1))$$
$$\leq 1 - \frac{\exp(-\mathrm{KL}(\mathsf{Q}(S \mid v_{j_x} = 1)\|\mathsf{Q}(S \mid v_{j_x} = -1)))}{2}. \tag{16}$$

Combining Eqs. (14)-(16) we get that

$$\mathfrak{R}_{\mathcal{V}}$$
$$\geq \frac{1}{2} - \frac{1}{2}\int_0^1 \mathsf{P}_{\mathsf{test}}(x)\left|\phi\left(x - \frac{j_x + 1/2}{K}\right)\right| \, \mathrm{d}x$$
$$+ \frac{1}{4}\int_0^1 \mathsf{P}_{\mathsf{test}}(x)\left|\phi\left(x - \frac{j_x + 1/2}{K}\right)\right| \exp(-\mathrm{KL}(\mathsf{Q}(S \mid v_{j_x} = 1)\|\mathsf{Q}(S \mid v_{j_x} = -1))) \, \mathrm{d}x. \tag{17}$$

**Upper bound on $\mathfrak{B}_{\mathcal{V}}$:**   The Bayes error is

$$\mathfrak{B}_{\mathcal{V}} = \mathbb{E}_V[R(f^\star(\mathsf{P}_V); \mathsf{P}_V)]$$
$$= \mathbb{E}_V\left[\inf_f \mathbb{E}_{(x,y)\sim\mathsf{P}_{v,\mathsf{test}}}\mathbf{1}(f(x) \neq y)\right]$$
$$= \mathbb{E}_V\left[\inf_f \int_{x=0}^1 \sum_{y\in\{-1,1\}} \mathsf{P}_{\mathsf{test}}(x)\mathsf{P}_{V,\mathsf{test}}(y \mid x)\mathbf{1}(f(x) = -y)\right]$$
$$= \mathbb{E}_V\left[\int_{x=0}^1 \mathsf{P}_{\mathsf{test}}(x) \min_{y\in\{-1,1\}} \mathsf{P}_{V,\mathsf{test}}(y \mid x)\right]$$
$$\stackrel{(i)}{=} \mathbb{E}_V\left[\frac{1}{2}\left(1 - \int_{x=0}^1 \mathsf{P}_{\mathsf{test}}(x)|\mathsf{P}_{V,\mathsf{test}}(1 \mid x) - \mathsf{P}_{V,\mathsf{test}}(-1 \mid x)| \, \mathrm{d}x\right)\right]$$
$$\stackrel{(ii)}{=} \mathbb{E}_V\left[\frac{1}{2}\left(1 - \int_{x=0}^1 \mathsf{P}_{\mathsf{test}}(x)\left|\phi\left(x - \frac{j_x + 1/2}{K}\right)\right| \, \mathrm{d}x\right)\right]$$
$$= \frac{1}{2} - \frac{1}{2}\int_{x=0}^1 \mathsf{P}_{\mathsf{test}}(x)\left|\phi\left(x - \frac{j_x + 1/2}{K}\right)\right| \, \mathrm{d}x, \tag{18}$$

where $(i)$ follows since $\mathsf{P}_v(1 \mid x) = 1 - \mathsf{P}_v(-1 \mid x)$ and $\min\{s, 1 - s\} = (1 - |1 - 2s|)/2$ for all $s \in [0, 1]$, and $(ii)$ follows by our construction of $\mathsf{P}_v$ above along with the fact that $\mathsf{P}_v(1 \mid x) = 1 - \mathsf{P}_v(-1 \mid x)$.

**Putting things together:**   Combining Eqs. (17) and (18) allows us to conclude that

Minimax Excess Risk$(\mathcal{P}_{\mathsf{GS}}(\tau))$

$$\geq \frac{1}{4}\int_0^1 \mathsf{P}_{\mathsf{test}}(x)\left|\phi\left(x - \frac{j_x + 1/2}{K}\right)\right| \exp(-\mathrm{KL}(\mathsf{Q}(S \mid v_{j_x} = 1)\|\mathsf{Q}(S \mid v_{j_x} = -1))) \, \mathrm{d}x$$
$$= \frac{1}{4}\sum_{j=1}^K \int_{\frac{j-1}{K}}^{\frac{j}{K}} \mathsf{P}_{\mathsf{test}}(x)\left|\phi\left(x - \frac{j + 1/2}{K}\right)\right| \exp(-\mathrm{KL}(\mathsf{Q}(S \mid v_j = 1)\|\mathsf{Q}(S \mid v_j = -1))) \, \mathrm{d}x$$
$$= \frac{1}{4}\sum_{j=1}^K \exp(-\mathrm{KL}(\mathsf{Q}(S \mid v_j = 1)\|\mathsf{Q}(S \mid v_j = -1)))\left[\int_{\frac{j-1}{K}}^{\frac{j}{K}} \mathsf{P}_{\mathsf{test}}(x)\left|\phi\left(x - \frac{j + 1/2}{K}\right)\right| \, \mathrm{d}x\right]$$
$$\stackrel{(i)}{=} \frac{1}{32K^2}\sum_{j=1}^K \exp(-\mathrm{KL}(\mathsf{Q}(S \mid v_j = 1)\|\mathsf{Q}(S \mid v_j = -1))),$$

where $(i)$ follows by using Lemma A.2 along with the fact that $\mathsf{P}_{\mathsf{test}}(x) = 1$ in our construction to show that the integral in the square brackets is equal to $1/8K^2$. This proves the result. $\qquad\square$

The next lemma upper bounds the KL divergence between $Q(S \mid v_j = 1)$ and $Q(S \mid v_j = -1)$ for each $j \in \{1, \ldots, K\}$. It shows that the KL divergence between these two posteriors is larger when the expected number of samples in that bin is larger.

**Lemma C.2.** *Suppose that $v$ is drawn uniformly from the set $\{-1, 1\}^K$, and that $S \mid v$ is drawn from $\mathsf{P}_{v,\mathsf{maj}}^{n_{\mathsf{maj}}} \times \mathsf{P}_{v,\mathsf{min}}^{n_{\mathsf{min}}}$. Then for any $j \in \{1, \ldots, K/2\}$ and any $\tau \in [0, 1]$,*

$$\mathrm{KL}(Q(S \mid v_j = 1) \| Q(S \mid v_j = -1)) \leq \frac{n_{\mathsf{maj}}(2 - \tau) + n_{\mathsf{min}}\tau}{3K^3},$$

*and for any $j \in \{K/2 + 1, \ldots, K\}$*

$$\mathrm{KL}(Q(S \mid v_j = 1) \| Q(S \mid v_j = -1)) \leq \frac{n_{\mathsf{maj}}\tau + n_{\mathsf{min}}(2 - \tau)}{3K^3}.$$

*Proof.* Let us consider the case when $j = 1$. The bound for all other $j \in \{2, \ldots, K\}$ shall follow analogously.

Given samples $S$, let $S = (S_1, \bar{S}_1)$ be a partition where $S_1$ are the samples that fall in the interval $I_1$, and $\bar{S}_1$ be the other samples. Similarly, given a vector $v \in \{-1, 1\}$, let $v = (v_1, \bar{v}_1)$, where $v_1$ is the first component and $\bar{v}_1$ denotes the other components $(2, \ldots, K)$ of $v$.

First, we will show that

$$Q(S \mid v_1) = Q(S_1 \mid v_1)Q(\bar{S}_1).$$

To see this, observe that

$$Q(S \mid v_1) = Q((S_1, \bar{S}_1) \mid v_1) = Q(S_1 \mid v_1)Q(\bar{S}_1 \mid v_1, S_1).$$

Further, if $v$ is chosen uniformly over the hypercube $\{-1, 1\}^K$, then

$$\begin{aligned}
Q(\bar{S}_1 \mid v_1, S_1) &= \sum_{\bar{v}_1} Q(\bar{S}_1, \bar{v}_1 \mid v_1, S_1) \\
&= \sum_{\bar{v}_1} Q(\bar{S}_1 \mid v_1, \bar{v}_1, S_1)Q(\bar{v}_1 \mid v_1, S_1) \\
&\overset{(i)}{=} \sum_{\bar{v}_1} Q(\bar{S}_1 \mid v_1, \bar{v}_1, S_1)Q(\bar{v}_1) \\
&\overset{(ii)}{=} \sum_{\bar{v}_1} Q(\bar{S}_1 \mid v_1, \bar{v}_1)Q(\bar{v}_1) \\
&\overset{(iii)}{=} \sum_{\bar{v}_1} Q(\bar{S}_1 \mid \bar{v}_1)Q(\bar{v}_1) \\
&= Q(\bar{S}_1),
\end{aligned}$$

where $(i)$ follows since by Bayes' rule

$$\begin{aligned}
Q(\bar{v}_1 \mid v_1, S_1) &= \frac{Q(\bar{v}_1 \mid v_1)Q(S_1 \mid v_1, \bar{v}_1)}{Q(S_1 \mid v_1)} \\
&= \frac{Q(\bar{v}_1)Q(S_1 \mid v_1, \bar{v}_1)}{Q(S_1 \mid v_1)} && \text{(since } \bar{v}_1 \text{ is independent of } v_1\text{)} \\
&= \frac{Q(\bar{v}_1)Q(S_1 \mid v_1)}{Q(S_1 \mid v_1)} = Q(\bar{v}_1) && \text{(the samples in } S_1 \text{ depend only on } v_1\text{).}
\end{aligned}$$

Inequality $(ii)$ follows since the samples are drawn independently given $v = (v_1, \bar{v}_1)$. Finally, $(iii)$ follows since $\bar{S}_1$ (the samples that lie outside the interval $I_1$) only depend on $\bar{v}_1$ since the marginal distribution of $x$ is independent of $v$ and the distribution of $y \mid x$ depends only on the value of $v$ corresponding to the interval in which $x$ lies.

Thus since, $\mathsf{Q}(S \mid v_1) = \mathsf{Q}(S_1 \mid v_1)\mathsf{Q}(\bar{S}_1)$ we have that

$$\mathrm{KL}(\mathsf{Q}(S \mid v_1 = 1)\|\mathsf{Q}(S \mid v_1 = -1)) = \mathrm{KL}(\mathsf{Q}(S_1 \mid v_1 = 1)\|\mathsf{Q}(S_1 \mid v_1 = -1)). \tag{19}$$

To bound this KL divergence, let us condition of the number of samples in $S_1$ from group $a$, (the majority group) $n_{1,a}$ and the number of samples from group $b$ (the minority group), $n_{1,b}$. Now since $n_{1,a}$ and $n_{1,b}$ are independent of $v_1$ (which only affects the labels) we have that,

$$
\begin{aligned}
\mathsf{Q}(S_1 \mid v_1) &= \sum_{n_{1,a},n_{1,b}} \mathsf{Q}(n_{1,a}, n_{1,b} \mid v_1)\mathsf{Q}(S_1 \mid v_1, n_{1,a}, n_{1,b}) \\
&= \sum_{n_{1,a},n_{1,b}} \mathsf{Q}(n_{1,a}, n_{1,b})\mathsf{Q}(S_1 \mid v_1, n_{1,a}, n_{1,b}) \\
&= \mathbb{E}_{n_{1,a},n_{1,b}} \left[ \mathsf{Q}(S_1 \mid v_1, n_{1,a}, n_{1,b}) \right].
\end{aligned}
$$

Therefore, by the joint convexity of the KL-divergence and by Jensen's inequality we have that,

$$
\begin{aligned}
&\mathrm{KL}(\mathsf{Q}(S_1 \mid v_1 = 1)\|\mathsf{Q}(S_1 \mid v_1 = -1)) \\
&\qquad \leq \mathbb{E}_{n_{1,a},n_{1,b}} \left[ \mathrm{KL}(\mathsf{Q}(S_1 \mid v_1 = 1, n_{1,a}, n_{1,b})\|\mathsf{Q}(S_1 \mid v_1 = -1, n_{1,a}, n_{1,b})) \right]. \tag{20}
\end{aligned}
$$

Now conditioned on $v_1, n_{1,a}$ and $n_{1,b}$, samples in $S_1$ are composed of 2 groups of samples $(S_{1,a}, S_{1,b})$. The samples in each group $(S_{1,a}, S_{1,b})$ are drawn independently from the distributions $\mathsf{P}_a(x \mid x \in I_1)\mathsf{P}_v(y \mid x)$ and $\mathsf{P}_b(x \mid x \in I_1)\mathsf{P}_v(y \mid x)$ respectively. Therefore,

$$
\begin{aligned}
&\mathrm{KL}(\mathsf{Q}(S_1 \mid v_1 = 1, n_{1,a}, n_{1,b})\|\mathsf{Q}(S_1 \mid v_1 = -1, n_{1,a}, n_{1,b})) \\
&\overset{(i)}{=} n_{1,a}\mathrm{KL}(\mathsf{P}_a(x \mid x \in I_1)\mathsf{P}_{v_1=1}(y \mid x)\|\mathsf{P}_a(x \mid x \in I_1)\mathsf{P}_{v_1=-1}(y \mid x)) \\
&\qquad + n_{1,b}\mathrm{KL}(\mathsf{P}_b(x \mid x \in I_1)\mathsf{P}_{v_1=1}(y \mid x)\|\mathsf{P}_b(x \mid x \in I_1)\mathsf{P}_{v_1=-1}(y \mid x)) \\
&\overset{(ii)}{=} (n_{1,a} + n_{1,b})\mathbb{E}_{x\sim\mathsf{Unif}(I_1)} \left[ \mathrm{KL}(\mathsf{P}_{v_1=1}(y \mid x)\|\mathsf{P}_{v_1=-1}(y \mid x)) \right] \\
&\overset{(iii)}{=} \frac{n_{1,a} + n_{1,b}}{2}\mathbb{E}_{x\sim\mathsf{Unif}(I_1)} \left[ \sum_{y\in\{-1,1\}} \left( 1 + y\phi\left( x - \frac{1}{2K} \right) \right) \log\left( \frac{\left( 1 + y\phi\left( x - \frac{1}{2K} \right) \right)}{\left( 1 + y\phi\left( x - \frac{1}{2K} \right) \right)} \right) \right] \\
&= \frac{n_{1,a} + n_{1,b}}{2} \sum_{y\in\{-1,1\}} \mathbb{E}_{x\sim\mathsf{Unif}(I_1)} \left[ \left( 1 + y\phi\left( x - \frac{1}{2K} \right) \right) \log\left( \frac{\left( 1 + y\phi\left( x - \frac{1}{2K} \right) \right)}{\left( 1 + y\phi\left( x - \frac{1}{2K} \right) \right)} \right) \right] \\
&= \frac{n_{1,a} + n_{1,b}}{2K} \sum_{y\in\{-1,1\}} \int_{x=0}^{\frac{1}{K}} \left[ \left( 1 + y\phi\left( x - \frac{1}{2K} \right) \right) \log\left( \frac{\left( 1 + y\phi\left( x - \frac{1}{2K} \right) \right)}{\left( 1 + y\phi\left( x - \frac{1}{2K} \right) \right)} \right) \right] \mathrm{d}x \\
&\overset{(iv)}{\leq} \frac{n_{1,a} + n_{1,b}}{3K^2}, \tag{21}
\end{aligned}
$$

where in $(i)$ we let $\mathsf{P}_{v_1}$ denote the conditional distribution of $y$ for $x \in I_1$ given $v_1$, $(ii)$ follows since both $\mathsf{P}_a$ and $\mathsf{P}_b$ are constant in the interval, $(iii)$ follows by our construction of $\mathsf{P}_v$ above, and finally $(iv)$ follows by invoking Lemma A.3 that ensures that the integral is bounded by $1/3K^2$.

Using this bound in Eq. (20), along with Eq. (19) we get that

$$\mathrm{KL}(\mathsf{Q}(S \mid v_1 = 1)\|\mathsf{Q}(S \mid v_1 = -1)) \leq \frac{\mathbb{E}\left[ n_{1,a} + n_{2,b} \right]}{3K^2}.$$

Now there are $n_{\mathsf{maj}}$ samples from group $a$ in $S$ and $n_{\mathsf{min}}$ samples from group $b$. Therefore,

$$
\begin{aligned}
\mathbb{E}\left[ n_{1,a} \right] &= n_{\mathsf{maj}}\mathsf{P}_a(x \in I_1) = \frac{n_{\mathsf{maj}}(2 - \tau)}{K}, \\
\mathbb{E}\left[ n_{1,b} \right] &= n_{\mathsf{min}}\mathsf{P}_b(x \in I_1) = \frac{n_{\mathsf{min}}\tau}{K}.
\end{aligned}
$$

Plugging this bound into Eq. (21) completes the proof by the first interval. An identical argument holds for $j \in \{2, \ldots, K/2\}$. For $j \in \{K/2+1, \ldots, K\}$ the only change is that

$$\mathbb{E}\left[n_{j,a}\right] = n_{\mathsf{maj}}\mathsf{P}_a(x \in I_j) = \frac{n_{\mathsf{maj}}\tau}{K},$$

$$\mathbb{E}\left[n_{j,b}\right] = n_{\mathsf{min}}\mathsf{P}_b(x \in I_j) = \frac{n_{\mathsf{min}}(2-\tau)}{K}.$$

$\square$

Next, we combine the previous two lemmas to establish our stated lower bound. We first restate it here.

**Theorem 4.2.** *Consider the group shift setting described in Section 3.2.2. Given any overlap* $\tau \in [0, 1]$ *recall that* $\mathcal{P}_{\mathsf{GS}}(\tau)$ *is the class of distributions such that* $\mathsf{Overlap}(\mathsf{P}_{\mathsf{maj}}, \mathsf{P}_{\mathsf{min}}) \geq \tau$. *The minimax excess risk in this setting is lower bounded as follows:*

$$\mathsf{Minimax\ Excess\ Risk}(\mathcal{P}_{\mathsf{GS}}(\tau)) = \inf_{\mathcal{A}} \sup_{(\mathsf{P}_{\mathsf{maj}}, \mathsf{P}_{\mathsf{min}}) \in \mathcal{P}_{\mathsf{GS}}(\tau)} \mathsf{Excess\ Risk}[\mathcal{A}; (\mathsf{P}_{\mathsf{maj}}, \mathsf{P}_{\mathsf{min}})]$$

$$\geq \frac{1}{200(n_{\mathsf{min}} \cdot (2-\tau) + n_{\mathsf{maj}} \cdot \tau)^{1/3}} \geq \frac{1}{200 n_{\mathsf{min}}^{1/3}(\rho \cdot \tau + 2)^{1/3}}, \tag{4}$$

*where* $\rho = n_{\mathsf{maj}}/n_{\mathsf{min}} > 1$.

*Proof.* First, by Lemma C.1 we know that

$$\mathsf{Minimax\ Excess\ Risk}(\mathcal{P}_{\mathsf{GS}}(\tau)) \geq \frac{1}{32K^2} \sum_{j=1}^{K} \exp(-\mathsf{KL}(\mathsf{Q}(S \mid v_j = 1)\|\mathsf{Q}(S \mid v_j = -1))).$$

Next, by invoking the bound on the KL divergences in the equation above by Lemma C.2 we get that

$$\mathsf{Minimax\ Excess\ Risk}(\mathcal{P}_{\mathsf{GS}}(\tau))$$

$$\geq \frac{1}{64K}\left[\exp\left(-\frac{n_{\mathsf{maj}}(2-\tau) + n_{\mathsf{min}}\tau}{3K^3}\right) + \exp\left(-\frac{n_{\mathsf{min}}(2-\tau) + n_{\mathsf{maj}}\tau}{3K^3}\right)\right]$$

$$\geq \frac{1}{64K}\left[\exp\left(-\frac{n_{\mathsf{min}}(2-\tau) + n_{\mathsf{maj}}\tau}{3K^3}\right)\right]$$

Setting $K = \lceil(n_{\mathsf{min}}(2-\tau) + n_{\mathsf{maj}}\tau)^{1/3}\rceil$ and recalling that $\tau \leq 1$ we get that

$$\mathsf{Minimax\ Excess\ Risk}(\mathcal{P}_{\mathsf{GS}}(\tau))$$

$$\geq \frac{1}{64\lceil(n_{\mathsf{min}}(2-\tau) + n_{\mathsf{maj}}\tau)^{1/3}\rceil}\left[\exp\left(-\frac{n_{\mathsf{min}}(2-\tau) + n_{\mathsf{maj}}\tau}{3\lceil(n_{\mathsf{min}}(2-\tau) + n_{\mathsf{maj}}\tau)^{1/3}\rceil^3}\right)\right]$$

$$\overset{(i)}{\geq} \frac{\exp(-1/3)}{64}\frac{(n_{\mathsf{min}}(2-\tau) + n_{\mathsf{maj}}\tau)^{1/3}}{\lceil(n_{\mathsf{min}}(2-\tau) + n_{\mathsf{maj}}\tau)^{1/3}\rceil}\frac{1}{(n_{\mathsf{min}}(2-\tau) + n_{\mathsf{maj}}\tau)^{1/3}}$$

$$\overset{(ii)}{\geq} \frac{0.7\exp(-1/3)}{64}\frac{1}{(n_{\mathsf{min}}(2-\tau) + n_{\mathsf{maj}}\tau)^{1/3}}$$

$$\geq \frac{1}{200}\frac{1}{(n_{\mathsf{min}}(2-\tau) + n_{\mathsf{maj}}\tau)^{1/3}},$$

where $(i)$ follows since $n_{\mathsf{min}}(2-\tau) + n_{\mathsf{maj}}\tau/\lceil(n_{\mathsf{min}}(2-\tau) + n_{\mathsf{maj}}\tau)^{1/3}\rceil^3 \leq 1$, and $(ii)$ follows since $0 \leq \tau \leq 1$ and $n_{\mathsf{min}} \geq 1$ and hence $\frac{(n_{\mathsf{min}}(2-\tau)+n_{\mathsf{maj}}\tau)^{1/3}}{\lceil(n_{\mathsf{min}}(2-\tau)+n_{\mathsf{maj}}\tau)^{1/3}\rceil} \geq 0.7$. $\square$

## C.2 Proof of Theorem 5.2

In this section, we derive an upper bound on the excess risk of the undersampled binning estimator $\mathcal{A}_{\mathsf{USB}}$ (Eq. (5)). Recall that given a dataset $\mathcal{S}$ this estimator first calculates the undersampled dataset $\mathcal{S}_{\mathsf{US}}$, where the number of points from the minority group ($n_{\mathsf{min}}$) is equal to the number of points from the majority group ($n_{\mathsf{min}}$), and the size of the dataset is $2n_{\mathsf{min}}$. Throughout this section, $(\mathsf{P}_{\mathsf{maj}}, \mathsf{P}_{\mathsf{min}})$ shall be an arbitrary element of $\mathcal{P}_{\mathsf{GS}}(\tau)$ for any $\tau \in [0,1]$. In this section, whenever we shall often denote $\mathsf{Excess\ Risk}(\mathcal{A}; (\mathsf{P}_{\mathsf{maj}}, \mathsf{P}_{\mathsf{min}}))$ by simply $\mathsf{Excess\ Risk}(\mathcal{A})$.

Before we proceed, we introduce some additional notation. For any $j \in \{1, \ldots, K\}$ and $I_j = [\frac{j-1}{K}, \frac{j}{K}]$ let

$$q_{j,1} := \mathsf{P}_{\mathsf{test}}(y = 1 \mid x \in I_j) = \int_{x \in I_j} \mathsf{P}(y = 1 \mid x)\mathsf{P}_{\mathsf{test}}(x \mid x \in I_j)\, \mathrm{d}x, \tag{22a}$$

$$q_{j,1} := \mathsf{P}_{\mathsf{test}}(y = 1 \mid x \in I_j) = \int_{x \in I_j} \mathsf{P}(y = 1 \mid x)\mathsf{P}_{\mathsf{test}}(x \mid x \in I_j)\, \mathrm{d}x. \tag{22b}$$

For the undersampled binning estimator $\mathcal{A}_{\mathsf{USB}}$ (defined above in Eq. (5)), define the *excess risk in an interval* $I_j$ as follows:

$$R_j(\mathcal{A}_{\mathsf{USB}}^{\mathcal{S}}) := p\left(y = -\mathcal{A}_j^{\mathcal{S}} \mid x \in I_j\right) - \min\left\{\mathsf{P}_{\mathsf{test}}(y = 1 \mid x \in I_j), \mathsf{P}_{\mathsf{test}}(y = -1 \mid x \in I_j)\right\}$$
$$= q_{j,-\mathcal{A}_j^{\mathcal{S}}} - \min\{q_{j,1}, q_{j,-1}\}.$$

The proof of the upper bound shall proceed in steps. First, in Lemma C.3 we will show that the excess risk is equal to sum the excess risk over the intervals up to a factor of $2/K$ on account of the distribution being 1-Lipschitz. Next, in Lemma C.4 we upper bound the risk over each interval. We put these two together and to upper bound the risk.

**Lemma C.3.** *The expected excess risk of undersampled binning estimator $\mathcal{A}_{\mathsf{USB}}$ can be decomposed as follows*

$$\mathsf{Excess\ Risk}(\mathcal{A}_{\mathsf{USB}}) \leq \sum_{j=0}^{K-1} \mathbb{E}_{\mathcal{S} \sim \mathsf{P}_{\mathsf{maj}}^{n_{\mathsf{maj}}} \times \mathsf{P}_{\mathsf{min}}^{n_{\mathsf{min}}}}\left[R_j(\mathcal{A}_{\mathsf{USB}}^{\mathcal{S}})\right] \cdot \mathsf{P}_{\mathsf{test}}(I_j) + \frac{2}{K},$$

*where $\mathsf{P}_{\mathsf{test}}(I_j) := \int_{x \in I_j} \mathsf{P}_{\mathsf{test}}(x)\, \mathrm{d}x$.*

*Proof.* Recall that by definition, the expected excess risk is

$$\mathbb{E}_{\mathcal{S} \sim \mathsf{P}_{\mathsf{maj}}^{n_{\mathsf{maj}}} \times \mathsf{P}_{\mathsf{min}}^{n_{\mathsf{min}}}}\left[R(\mathcal{A}^{\mathcal{S}}; \mathsf{P}_{\mathsf{test}}) - R(f^{\star}; \mathsf{P}_{\mathsf{test}})\right].$$

Let us first decompose the Bayes risk $R(f^{\star})$,

$$R(f^{\star}) = \inf_f \mathbb{E}_{(x,y) \sim \mathsf{P}_{\mathsf{test}}}\left[\mathbf{1}(f(x) \neq y)\right]$$
$$= \inf_f \int_{x=0}^{1} \sum_{y \in \{-1,1\}} \mathbf{1}(f(x) \neq y)\mathsf{P}_{\mathsf{test}}(y \mid x)\mathsf{P}_{\mathsf{test}}(x)\, \mathrm{d}x$$
$$= \int_{x=0}^{1} \inf_{f(x) \in \{-1,1\}} \sum_{y \in \{-1,1\}} \mathbf{1}(f(x) \neq y)\mathsf{P}_{\mathsf{test}}(y \mid x)\mathsf{P}_{\mathsf{test}}(x)\, \mathrm{d}x$$
$$= \int_{x=0}^{1} \inf_{f(x) \in \{-1,1\}} \mathsf{P}_{\mathsf{test}}(y = -f(x) \mid x)\mathsf{P}_{\mathsf{test}}(x)\, \mathrm{d}x$$
$$= \int_{x=0}^{1} \min\left\{\mathsf{P}_{\mathsf{test}}(y = 1 \mid x), \mathsf{P}_{\mathsf{test}}(y = -1 \mid x)\right\}\mathsf{P}_{\mathsf{test}}(x)\, \mathrm{d}x. \tag{23}$$

The risk of the undersampled binning algorithm $\mathcal{A}_{\mathsf{USB}}$ is given by

$$R(\mathcal{A}_{\mathsf{USB}}^{\mathcal{S}}) = \int_{x=0}^{1} \sum_{y \in \{-1,1\}} \mathbf{1}(\mathcal{A}_{\mathsf{USB}}^{\mathcal{S}}(x) \neq y) \mathsf{P}_{\mathsf{test}}(y \mid x) \mathsf{P}_{\mathsf{test}}(x) \, \mathrm{d}x$$

$$= \int_{x=0}^{1} \mathsf{P}_{\mathsf{test}}(y = -\mathcal{A}_{\mathsf{USB}}^{\mathcal{S}}(x) \mid x) \mathsf{P}_{\mathsf{test}}(x) \, \mathrm{d}x.$$

Next, recall that the undersampled binning estimator is constant over the intervals $I_j$ for $j \in \{1, \ldots, K\}$ where it takes the value $\mathcal{A}_j^{\mathcal{S}}$ (to ease notation let us simply denote it by $\mathcal{A}_j$ below), and therefore

$$R(\mathcal{A}_{\mathsf{USB}}^{\mathcal{S}}) = \sum_{j=0}^{K-1} \int_{x \in I_j} \mathsf{P}_{\mathsf{test}}(y = -\mathcal{A}_j | x) \mathsf{P}_{\mathsf{test}}(x) \, \mathrm{d}x.$$

This combined with Eq. (23) tells us that

$$R(\mathcal{A}_{\mathsf{USB}}^{\mathcal{S}}) - R(f^\star)$$
$$= \sum_{j=0}^{K-1} \int_{x \in I_j} \left( \mathsf{P}_{\mathsf{test}}(y = -\mathcal{A}_j | x) - \min \{ \mathsf{P}_{\mathsf{test}}(y = 1 \mid x), \mathsf{P}_{\mathsf{test}}(y = -1 \mid x) \} \right) \mathsf{P}_{\mathsf{test}}(x) \, \mathrm{d}x. \qquad (24)$$

Recall the definition of $q_{j,1}$ and $q_{j,-1}$ from Eqs. (22a)-(22b) above. For any $x \in I_j = [\frac{j-1}{K}, \frac{j}{K}]$, $|\mathsf{P}_{\mathsf{test}}(y \mid x) - q_{j,y}| \leq 1/K$, since the distribution $\mathsf{P}_{\mathsf{test}}(y \mid x)$ is 1-Lipschitz and $q_{j,y}$ is its conditional mean. Therefore,

$$R(\mathcal{A}_{\mathsf{USB}}^{\mathcal{S}}) - R(f^\star)$$
$$\leq \sum_{j=0}^{K-1} \int_{x \in I_j} \left( q_{j,-\mathcal{A}_j} - \min \{ q_{j,1}, q_{j,-1} \} \right) \mathsf{P}_{\mathsf{test}}(x) \, \mathrm{d}x + \frac{2}{K} \sum_{j=0}^{K-1} \int_{x \in I_j} \mathsf{P}_{\mathsf{test}}(x) \, \mathrm{d}x$$
$$= \sum_{j=0}^{K-1} \int_{x \in I_j} R_j(\mathcal{A}_{\mathsf{USB}}^{\mathcal{S}}) \mathsf{P}_{\mathsf{test}}(x) \, \mathrm{d}x + \frac{2}{K}.$$

Taking expectation over the training samples $\mathcal{S}$ (where $n_{\mathsf{min}}$ samples are drawn independently from $\mathsf{P}_{\mathsf{min}}$ and $n_{\mathsf{maj}}$ samples are drawn independently from $\mathsf{P}_{\mathsf{maj}}$) concludes the proof. $\qquad \square$

Next we provide an upper bound on the expected excess risk is an interval $R_j(\mathcal{A}_{\mathsf{USB}}^{\mathcal{S}})$.

**Lemma C.4.** *For any $j \in \{1, \ldots, K\}$ with $I_j = [\frac{j-1}{K}, \frac{j}{K}]$,*

$$\mathbb{E}_{\mathcal{S} \sim \mathsf{P}_{\mathsf{maj}}^{n_{\mathsf{maj}}} \times \mathsf{P}_{\mathsf{min}}^{n_{\mathsf{min}}}} \left[ R_j(\mathcal{A}_{\mathsf{USB}}^{\mathcal{S}}) \right] \leq \frac{c}{\sqrt{n_{\mathsf{min}} \mathsf{P}_{\mathsf{test}}(I_j)}} + \frac{c}{K},$$

*where $c$ is an absolute constant, and $\mathsf{P}_{\mathsf{test}}(I_j) := \int_{x \in I_j} \mathsf{P}_{\mathsf{test}}(x) \, \mathrm{d}x$.*

*Proof.* Consider an arbitrary bucket $j \in \{1, \ldots, K\}$.

Let us introduce some notation that shall be useful in the remainder of the proof. Analogous to $q_{j,1}$ and $q_{j,-1}$ defined above (see Eqs. (22a)-(22b)), define $q_{j,1}^a$ and $q_{j,1}^b$ as follows:

$$q_{j,1}^a := \mathsf{P}_a(y = 1 \mid x \in I_j) = \int_{x \in I_j} \mathsf{P}(y = 1 \mid x) \mathsf{P}_a(x \mid x \in I_j) \, \mathrm{d}x, \qquad (25a)$$

$$q_{j,1}^b := \mathsf{P}_b(y = 1 \mid x \in I_j) = \int_{x \in I_j} \mathsf{P}(y = 1 \mid x) \mathsf{P}_b(x \mid x \in I_j) \, \mathrm{d}x. \qquad (25b)$$

Essentially, $q_{j,1}^a$ is the probability that a sample is from group $a$ and has label 1, conditioned on the event that the sample falls in the interval $I_j$. Since

$$\mathsf{P}_{\mathsf{test}}(x \mid x \in I_j) = \frac{1}{2} \left[ \mathsf{P}_a(x \mid x \in I_j) + \mathsf{P}_b(x \mid x \in I_j) \right],$$

therefore

$$|q_{j,1} - q_{j,1}^a| = \left| \int_{x \in I_j} \mathsf{P}(y = 1 \mid x) \mathsf{P}_{\mathsf{test}}(x \mid x \in I_j) \, \mathrm{d}x - \int_{x \in I_j} \mathsf{P}(y = 1 \mid x) \mathsf{P}_a(x \mid x \in I_j) \, \mathrm{d}x \right|$$

$$\leq \frac{1}{K}. \tag{26}$$

This follows since $\mathsf{P}(y \mid x)$ is 1-Lipschitz and therefore can fluctuate by at most $1/K$ in the interval $I_j$. Of course the same bound also holds for $|q_{j,1} - q_{j,1}^b|$.

With this notation in place let us present a bound on the expected value of $R_j(\mathcal{A}_{\mathsf{USB}}^{\mathcal{S}})$. By definition

$$R_j(\mathcal{A}_{\mathsf{USB}}^{\mathcal{S}}) = q_{j,-\mathcal{A}_j^{\mathcal{S}}} - \min\{q_{j,1}, q_{j,-1}\}.$$

First, note that $q_{j,1} := \mathsf{P}_{\mathsf{test}}(y = 1 \mid x \in I_j) = 1 - q_{j,-1}$. Suppose that $q_{j,1} < 1/2$ and therefore $q_{j,-1} > 1/2$ (the same bound shall hold in the other case). In this case, risk is incurred only when $\mathcal{A}_j^{\mathcal{S}} = 1$. That is,

$$\mathbb{E}_{\mathcal{S} \sim \mathsf{P}_{\mathsf{maj}}^{n_{\mathsf{maj}}} \times \mathsf{P}_{\mathsf{min}}^{n_{\mathsf{min}}}} \left[ R_j(\mathcal{A}_{\mathsf{USB}}^{\mathcal{S}}) \right] = |q_{j,-1} - q_{j,1}| \Pr_{\mathcal{S}}[\mathcal{A}_j^{\mathcal{S}} = 1]$$

$$= |1 - 2q_{j,1}| \Pr_{\mathcal{S}}[\mathcal{A}_j^{\mathcal{S}} = 1]. \tag{27}$$

Now by the definition of the undersampled binning estimator (see Eq. (5)), $\mathcal{A}_j^{\mathcal{S}} = 1$ only when there are more samples in the interval $I_j$ with label 1 than $-1$. However, we can bound the probability of this happening since $q_{j,1}$ is smaller than $q_{j,-1}$.

Let $n_j$ be the number of samples in the undersampled sample set $\mathcal{S}_{\mathsf{US}}$ in the interval $I_j$. Let $n_{1,j}$ be the number of these samples with label 1, and $n_{-1,j} = n_j - n_{1,j}$ be the number of samples with label $-1$. Further, let $n_{a,j}$ be the number of samples in from group $a$ such that they fall in the interval $I_j$, and define $m_{b,j}$ analogously.

The probability of incurring risk is given by

$$\mathbb{P}[\mathcal{A}_j = 1] = \sum_{s=1}^{2n_{\mathsf{min}}} \mathbb{P}[\mathcal{A}_j = 1 \mid n_j = s] \mathbb{P}[n_j = s], \tag{28}$$

where the sum is up to $2n_{\mathsf{min}}$ since the size of the undersample dataset $|\mathcal{S}_{\mathsf{US}}|$ is equal to $2n_{\mathsf{min}}$.

Conditioned on the event that $n_j = s$ the probability of incurring risk is

$$\mathbb{P}\left[\mathcal{A}_j = 1 \mid n_j = s\right] = \mathbb{P}\left[m_{1,j} > n_{-1,j} \mid n_j = s\right] = \mathbb{P}\left[n_{1,j} > n_j/2 \mid n_j = s\right]$$

$$= \mathbb{P}\left[n_{1,j} > s/2 \mid n_j = s\right]. \tag{29}$$

Now, note that $n_j = n_{a,j} + n_{b,j}$. Thus continuing, we have that

$$\mathbb{P}\left[n_{1,j} > s/2 \mid n_j = s\right] = \sum_{s' \leq s} \mathbb{P}\left[n_{1,j} > s/2 \mid n_j = s, n_{b,j} = s'\right] \mathbb{P}[n_{b,j} = s']$$

$$= \sum_{s' \leq s} \mathbb{P}\left[n_{1,j} > s/2 \mid n_{a,j} = s - s', n_{b,j} = s'\right] \mathbb{P}[n_{b,j} = s'].$$

In light of this previous equation, we want to control the probability that the number of samples with label 1 in the interval $I_j$ conditioned on the event that the number of samples from group $a$ in this interval is $s - s'$ and the number of samples from group $b$ in this interval is $s'$. Recall that $q_{j,1}^a$ and $q_{j,1}^b$ the probabilities of the label of the sample being 1 conditioned the event that sample is in the interval $I_j$ when it is group $a$ and $b$ respectively. So we define the random variables:

$$z_a[s - s'] \sim \mathsf{Bin}(s - s', q_{j,1}^a), \quad z_b[s'] \sim \mathsf{Bin}(s', q_{j,1}^b), \quad z[s] \sim \mathsf{Bin}(s, \max\left\{q_{j,1}^a, q_{j,1}^b\right\}).$$

Then,

$$\mathbb{P}\left[n_{1,j} > s/2 \mid n_j = s\right]$$

$$= \sum_{s' \leq s} \mathbb{P}\left[n_{1,j} > s/2 \mid n_{j,a} = s - s', n_{j,b} = s'\right] \mathbb{P}[n_{j,b} = s']$$

$$= \sum_{s' \leq s} \mathbb{P}\left[z_a[s - s'] + z_b[s']) > s/2 \mid n_{a,j} = s - s', n_{b,j} = s'\right] \mathbb{P}[n_{b,j} = s']$$

$$\leq \sum_{s' \leq s} \mathbb{P}\left[z[s] > s/2 \mid n_{a,j} = s - s', n_{b,j} = s'\right] \mathbb{P}[n_{b,j} = s']$$

$$= \sum_{s' \leq s} \mathbb{P}\left[z[s] > s/2\right] \mathbb{P}[n_{b,j} = s']$$

$$= \mathbb{P}\left[z[s] > s/2\right]$$

$$\overset{(i)}{\leq} \exp\left(-\frac{s}{2}(1 - 2\max\{q_{j,1}^a, q_{j,1}^b\})^2\right), \tag{30}$$

where $(i)$ follows by invoking Hoeffding's inequality(Wainwright, 2019, Proposition 2.5). Combining this with Eqs. (28) and (29) we get that

$$\mathbb{P}[\mathcal{A}_j = 1] \leq \sum_{s=1}^{2n_{\min}} \exp\left(-\frac{s}{2}(1 - 2\max\{q_{j,1}^a, q_{j,1}^b\})^2\right) \mathbb{P}[n_j = s].$$

Now $n_j$, which is the number of samples that lands in the interval $I_j$ is equal to $n_{a,j} + n_{b,j}$. Now each of $n_{a,j}$ and $n_{b,j}$ (the number of samples in this interval from each of the groups) are random variables with distributions $\mathsf{Bin}(n_{\min}, \mathsf{P}_a(I_j))$ and $\mathsf{Bin}(n_{\min}, \mathsf{P}_b(I_j))$, where $\mathsf{P}_a(I_j) = \int_{x \in I_j} \mathsf{P}_a(x) \, \mathrm{d}x$ and $\mathsf{P}_b(I_j) = \int_{x \in I_j} \mathsf{P}_a(x) \, \mathrm{d}x$. Therefore, $n_j$ is distributed as a sum of two binomial distribution and is therefore Poisson binomially distributed (Wikipedia contributors, 2022). Using the formula for the moment generating function (MGF) of a Poisson binomially distributed random variable we infer that,

$$\mathbb{P}[\mathcal{A}_j = 1] \leq \left(1 - \mathsf{P}_a(I_j) + \mathsf{P}_a(I_j) \exp\left(-\frac{(1 - 2\max\{q_{j,1}^a, q_{j,1}^b\})^2}{2}\right)\right)^{n_{\min}} \times$$

$$\left(1 - \mathsf{P}_b(I_j) + \mathsf{P}_b(I_j) \exp\left(-\frac{(1 - 2\max\{q_{j,1}^a, q_{j,1}^b\})^2}{2}\right)\right)^{n_{\min}}.$$

Plugging this into Eq. (28) we get that,

$$\mathbb{E}_{\mathcal{S} \sim \mathsf{P}_{\mathsf{maj}}^{n_{\mathsf{maj}}} \times \mathsf{P}_{\mathsf{min}}^{n_{\mathsf{min}}}} \left[R_j(\mathcal{A}_{\mathsf{USB}}^{\mathcal{S}})\right]$$

$$\leq |1 - 2q_{j,1}| \left[1 - \mathsf{P}_a(I_j) + \mathsf{P}_a(I_j) \exp\left(-\frac{(1 - 2\max\{q_{j,1}^a, q_{j,1}^b\})^2}{2}\right)\right]^{n_{\min}} \times$$

$$\left[1 - \mathsf{P}_b(I_j) + \mathsf{P}_b(I_j) \exp\left(-\frac{(1 - 2\max\{q_{j,1}^a, q_{j,1}^b\})^2}{2}\right)\right]^{n_{\min}}$$

$$= |1 - 2q_{j,1}| \left[1 - \mathsf{P}_a(I_j) \left(1 - \exp\left(-\frac{(1 - 2\max\{q_{j,1}^a, q_{j,1}^b\})^2}{2}\right)\right)\right]^{n_{\min}} \times$$

$$\left[1 - \mathsf{P}_b(I_j) \left(1 - \exp\left(-\frac{(1 - 2\max\{q_{j,1}^a, q_{j,1}^b\})^2}{2}\right)\right)\right]^{n_{\min}}.$$

Since $|1 - 2\max\{q_{j,1}^a, q_{j,1}^b\}| \leq 1$,

$$1 - \exp\left(-\frac{(1 - 2\max\{q_{j,1}^a, q_{j,1}^b\})^2}{2}\right) \geq \frac{(1 - 2\max\{q_{j,1}^a, q_{j,1}^b\})^2}{4},$$

and therefore

$$
\mathbb{E}_{\mathcal{S} \sim \mathsf{P}_{\mathsf{maj}}^{n_{\mathsf{maj}}} \times \mathsf{P}_{\mathsf{min}}^{n_{\mathsf{min}}}} \left[ R_j(\mathcal{A}_{\mathsf{USB}}^{\mathcal{S}}) \right] \leq |1 - 2q_{j,1}| \left[ 1 - \mathsf{P}_a(I_j) \frac{(1 - 2\max\{q_{j,1}^a, q_{j,1}^b\})^2}{2} \right]^{n_{\mathsf{min}}} \times
$$

$$
\left[ 1 - \mathsf{P}_b(I_j) \frac{(1 - 2\max\{q_{j,1}^a, q_{j,1}^b\})^2}{2} \right]^{n_{\mathsf{min}}}
$$

$$
\overset{(i)}{\leq} |1 - 2q_{j,1}| \left[ 1 - \mathsf{P}_a(I_j) \frac{(1 - 2q_{j,1} - 2\gamma)^2}{2} \right]^{n_{\mathsf{min}}} \times
$$

$$
\left[ 1 - \mathsf{P}_b(I_j) \frac{(1 - 2q_{j,1} - 2\gamma)^2}{2} \right]^{n_{\mathsf{min}}}
$$

$$
\overset{(ii)}{\leq} |1 - 2q_{j,1}| \exp\left( -n_{\mathsf{min}}(\mathsf{P}_a(I_j) + \mathsf{P}_b(I_j)) \frac{(1 - 2q_{j,1} - 2\gamma)^2}{2} \right),
$$

where $(i)$ follows since $|\max\{q_{j,1}^a, q_{j,1}^b\} - q_{j,1}| \leq 1/K$ by Eq. (26) and $\gamma$ is such that $|\gamma| \leq 1/K$, and $(ii)$ follows since $(1 + z)^b \leq \exp(bz)$. Now the RHS above is maximized when $(1 - 2q_{j,1} - 2\gamma)^2 = \frac{c}{n_{\mathsf{min}}(\mathsf{P}_a(I_j) + \mathsf{P}_b(I_j))}$, for some constant $c$. Plugging this into the equation above we get that

$$
\mathbb{E}_{\mathcal{S} \sim \mathsf{P}_{\mathsf{maj}}^{n_{\mathsf{maj}}} \times \mathsf{P}_{\mathsf{min}}^{n_{\mathsf{min}}}} \left[ R_j(\mathcal{A}_{\mathsf{USB}}^{\mathcal{S}}) \right] \leq \frac{c'}{\sqrt{n_{\mathsf{min}}(\mathsf{P}_a(I_j) + \mathsf{P}_b(I_j))}} + c'|\gamma|
$$

$$
\leq \frac{c'}{\sqrt{n_{\mathsf{min}}(\mathsf{P}_a(I_j) + \mathsf{P}_b(I_j))}} + \frac{c'}{K}.
$$

Finally, noting that $\mathsf{P}_{\mathsf{test}}(I_j) = (\mathsf{P}_a(I_j) + \mathsf{P}_b(I_j))/2$ completes the proof. $\qquad\square$

By combining the previous two lemmas we can now prove our upper bound on the risk of the undersampled binning estimator. We begin by restating it.

**Theorem 5.2.** *Consider the group shift setting described in Section 3.2.2. For any overlap $\tau \in [0, 1]$ and for any $(\mathsf{P}_{\mathsf{maj}}, \mathsf{P}_{\mathsf{min}}) \in \mathcal{P}_{\mathsf{GS}}(\tau)$ the expected excess risk of the Undersampling Binning Estimator (Eq. (5)) with number of bins with $K = \lceil n_{\mathsf{min}}^{1/3} \rceil$ is*

$$
\mathsf{Excess\ Risk}[\mathcal{A}_{\mathsf{USB}}; (\mathsf{P}_{\mathsf{maj}}, \mathsf{P}_{\mathsf{min}})] = \mathbb{E}_{\mathcal{S} \sim \mathsf{P}_{\mathsf{maj}}^{n_{\mathsf{maj}}} \times \mathsf{P}_{\mathsf{min}}^{n_{\mathsf{min}}}} \left[ R(\mathcal{A}_{\mathsf{USB}}^{\mathcal{S}}; \mathsf{P}_{\mathsf{test}})) - R(f^\star; \mathsf{P}_{\mathsf{test}}) \right] \leq \frac{C}{n_{\mathsf{min}}^{1/3}}.
$$

*Proof.* First by Lemma C.3 we know that

$$
\mathsf{Excess\ Risk}[\mathcal{A}_{\mathsf{USB}}] \leq \sum_{j=0}^{K-1} \mathbb{E}_{\mathcal{S} \sim \mathsf{P}_{\mathsf{maj}}^{n_{\mathsf{maj}}} \times \mathsf{P}_{\mathsf{min}}^{n_{\mathsf{min}}}} \left[ R_j(\mathcal{A}_{\mathsf{USB}}^{\mathcal{S}}) \right] \cdot \mathsf{P}_{\mathsf{test}}(I_j) + \frac{2}{K}.
$$

Next by using the bound on $\mathbb{E}_{\mathcal{S} \sim \mathsf{P}_{\mathsf{maj}}^{n_{\mathsf{maj}}} \times \mathsf{P}_{\mathsf{min}}^{n_{\mathsf{min}}}} \left[ R_j(\mathcal{A}_{\mathsf{USB}}^{\mathcal{S}}) \right]$ established in Lemma C.4 we get that,

$$
\mathsf{Excess\ Risk}(\mathcal{A}_{\mathsf{USB}}) \leq c \sum_{j=0}^{K-1} \frac{1}{\sqrt{n_{\mathsf{min}}\mathsf{P}_{\mathsf{test}}(I_j)}} \mathsf{P}_{\mathsf{test}}(I_j) + \frac{c}{K}
$$

$$
= \frac{c}{\sqrt{n_{\mathsf{min}}}} \sum_{j=0}^{K-1} \sqrt{\mathsf{P}_{\mathsf{test}}(I_j)} + \frac{c}{K}
$$

$$
\overset{(i)}{\leq} \frac{c}{\sqrt{n_{\mathsf{min}}}} \sqrt{K} \sum_{j=0}^{K-1} \mathsf{P}_{\mathsf{test}}(I_j) + \frac{c}{K}
$$

$$
= c\sqrt{\frac{K}{n_{\mathsf{min}}}} + \frac{c}{K}.
$$

where $(i)$ follows since for any vector $z \in \mathbb{R}^K$, $\|z\|_1 \leq \sqrt{K}\|z\|_2$. Maximizing over $K$ yields the choice $K = \lceil n_{\mathsf{min}}^{1/3} \rceil$, completing the proof.

$\square$

## D Additional simulations

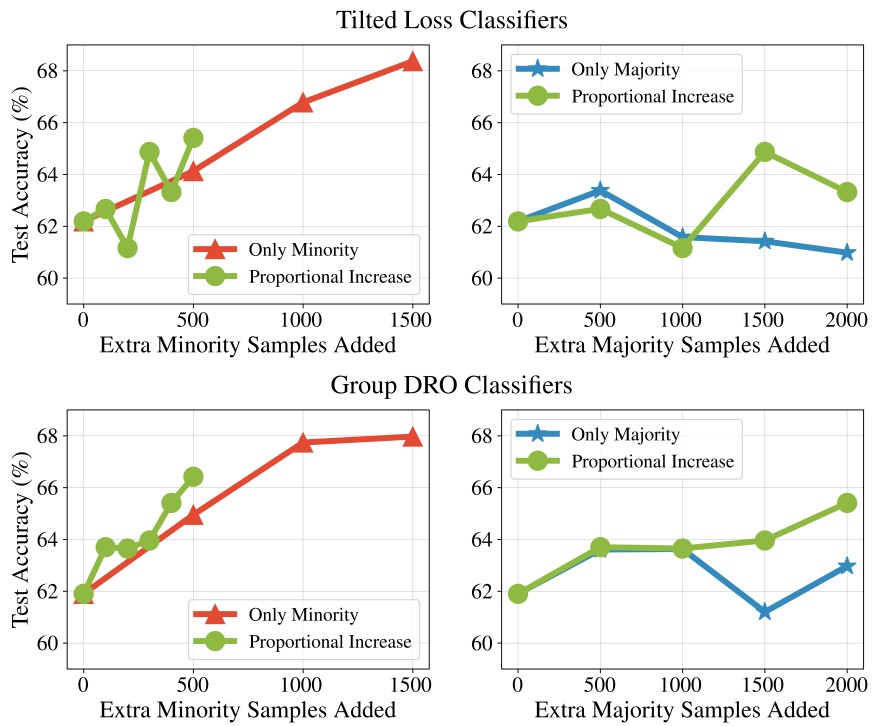

Figure 4: Convolutional neural network classifiers trained on the Imbalanced Binary CIFAR10 dataset with a 5:1 label imbalance. (Top) Models trained using the tilted loss (Li et al., 2020) with early stopping. (Bottom) Models trained using group-DRO (Sagawa et al., 2020) with early stopping. We report the average test accuracy calculated on a balanced test set over 5 random seeds. We start off with 2500 cat examples and 500 dog examples in the training dataset. We find similar trends to those obtained in Figure 2 even with these losses that are designed to optimize for the worst group accuracy.

## E Discussion about minimax lower bounds for cost-sensitive losses applied to the label shift setting

We add a more detailed discussion about applying minimax cost-sensitive losses to obtain a lower bound in the presence of label shift.

Assume that $\mathsf{P}_{\mathsf{maj}}$ is distribution of the covariates $x \mid y = 1$, and $\mathsf{P}_{\mathsf{min}}$ is the distribution of the covariates $x \mid y = -1$. The training samples are drawn from the distribution:

$$\mathsf{P}(x, y) = \mathsf{P}(y = 1)\mathsf{P}_{\mathsf{maj}} + \mathsf{P}(y = -1)\mathsf{P}_{\mathsf{min}},$$

where

$$\mathsf{P}(y = 1) = \frac{\rho}{1 + \rho} \quad \text{and} \quad \mathsf{P}(y = -1) = \frac{1}{1 + \rho}$$

for some imbalance ratio $\rho > 1$. On average the ratio between the number of points from the majority class to the number of points from the minority class is equal to $\rho$.

We set the cost of getting an incorrectly predicting the majority class label to be equal to

$$c_1 = \frac{1}{1+\rho}$$

and the cost of incorrectly predicting the minority class label to be equal to

$$c_{-1} = \frac{\rho}{1+\rho}.$$

Note that the costs $c_1 + c_{-1} = 1$ and that $c_1 < c_{-1}$.

The expected cost-sensitive loss is therefore equal to

$$
\begin{aligned}
\mathbb{E}_{(x,y)\sim\mathsf{P}}\left[c_y \mathbf{1}\left[f(x) \neq y\right]\right] &= \frac{\rho}{1+\rho}\mathbb{E}_{(x)\sim\mathsf{P}}\left[c_1 \mathbf{1}\left[f(x) \neq 1\right]\right] + \frac{1}{1+\rho}\mathbb{E}_{(x)\sim\mathsf{P}}\left[c_{-1}\mathbf{1}\left[f(x) \neq -1\right]\right] \\
&= \frac{\rho}{(1+\rho)^2}\mathbb{E}_{x\sim\mathsf{P}_{\mathsf{maj}}}\left[\mathbf{1}\left[f(x) \neq 1\right]\right] + \frac{\rho}{(1+\rho)^2}\mathbb{E}_{x\sim\mathsf{P}_{\mathsf{min}}}\left[\mathbf{1}\left[f(x) \neq -1\right]\right] \\
&= \frac{2\rho}{(1+\rho)^2}\mathbb{E}_{y\sim\mathsf{Unif}(-1,1),x\sim\mathsf{P}(x|y)}\left[\mathbf{1}\left[f(x) \neq y\right]\right].
\end{aligned}
$$

Now if we invoke the minimax lower bound (Kamalaruban & Williamson, 2018, Theorem 4) we get that

$$\min_f \max_P \frac{2\rho}{(1+\rho)^2}\mathbb{E}_{y\sim\mathsf{Unif}(-1,1),x\sim\mathsf{P}(x|y)}\left[\mathbf{1}\left[f(x) \neq y\right]\right] \geq \frac{C}{1+\rho}\min\left\{\sqrt{\frac{V}{(1+\rho)n}}, \frac{1}{1+\rho}\frac{V}{nh}\right\},$$

where the minimum over $f$ is over all measurable functions from the training data to binary labels, the maximum is over a data distribution that can be correctly classified with a classifier from a VC class with VC dimension at most $V$ and $h$ is the Massart noise margin. For more thorough definitions we urge the reader to see (Kamalaruban & Williamson, 2018). With this lower bound we get that

$$
\begin{aligned}
\min_f \max_P \mathbb{E}_{y\sim\mathsf{Unif}(-1,1),x\sim\mathsf{P}(x|y)}\left[\mathbf{1}\left[f(x) \neq y\right]\right] &\geq \frac{C(1+\rho)}{2\rho}\min\left\{\sqrt{\frac{V}{(1+\rho)n}}, \frac{1}{1+\rho}\frac{V}{nh}\right\} \\
&\geq \frac{C}{2}\min\left\{\sqrt{\frac{V}{(1+\rho)n}}, \frac{1}{1+\rho}\frac{V}{nh}\right\}.
\end{aligned}
$$

Therefore we find that this lower bound gets smaller as the imbalance ratio $\rho$ gets larger, predicting the wrong trend for the label shift problem.

## F Details about results in Table 1

In Table 1, we listed results regarding the performance of undersampled algorithms to others that are reported in the literature. Here we provide detailed references to these results.

**Label shift.** The results for label shift are from the paper by Cao et al. (2019). The results are reported in Table 2 of that paper. For Imb CIFAR 10 (step 10), the undersampling result corresponds to the entry CB RS from that table with accuracy 84.59% (error 15.41%), while the best method corresponds to the method LDAM-DRW with accuracy 87.81% (error 12.19%). For Imb CIFAR100 (step 10), the undersampling result again corresponds to CB RS with accuracy 53.08% (error 46.92%) while the best method corresponds to the method LDAM-DRW with accuracy 59.46% (error 40.54%).

**Group-covariate shift.** The results for the group-covariate shift are from Table 2 in Idrissi et al. (2022). For the CelebA dataset, the undersampled accuracy corresponds to the method SUBG and the best accuracy is for gDRO. For the Waterbirds dataset, the undersampled method is SUBG and the best competitor is RWG. For the MultiNLI dataset, the undersampled accuracy corresponds to the method SUBG and the best accuracy is for gDRO. Finally, for the CivilComments dataset, the undersampled method is SUBG and the best method is RWG.

# G   Experimental details for Figures 2 and 4

We construct our label shift dataset from the original CIFAR10 dataset. We create a binary classification task using the "cat" and "dog" classes. We use the official test examples as the balanced test set with 1000 cats and 1000 dogs. To form the initial train and validation sets, we use 2500 cat examples (half of the training set) and 500 dog examples, corresponding to a 5:1 label imbalance. We use 80% of those examples for training and the rest for validation. We are left with 2500 additional cat examples and 4500 dog examples from the original train set which we add into our training set to generate Figure 2.

We use the same convolutional neural network architecture as (Byrd & Lipton, 2019; Wang et al., 2022) with random initializations for this dataset. We train this model using SGD for 800 epochs with batchsize 64, a constant learning rate 0.001 and momentum 0.9. The importance weights used upweight the minority class samples in the training loss and validation loss is calculated to be $\frac{\text{\#Cat Train Examples}}{\text{\#Dog Train Examples}}$. We note that all of the experiments were performed on an internal cluster on 8 GPUs.

**VS loss:**   Given a dataset $\{x_i, y_i\}_{i=1}^n$, the VS loss (Kini et al., 2021) is defined as follows

$$\mathcal{L}_{\mathsf{VS}}(f) := \sum_{i=1}^n \log\left(1 + \exp\left(-\left(\frac{n_{g_i}}{n_{\max}}\right)^\gamma y_i f(x_i) - \frac{\tau n_{g_i}}{n}\right)\right),$$

where $g_i$ denotes the group label, $n_{g_i}$ corresponds to the number of samples from the group, $n_{\max}$ is the number of samples in the largest group and $n$ is the total number of samples. We set $\tau = 3$ and $\gamma = 0.3$, the best hyperparameters identified by Wang et al. (2022) on this dataset for this neural network architecture.

**Tilted loss:**   The tilted loss (Li et al., 2020) is defined as

$$\mathcal{L}_{\mathsf{Tilted}}(f) := \frac{1}{t} \log\left[\sum_{i=1}^n \exp\left(t\ell(y_i f(x_i))\right)\right],$$

where we take $\ell$ to be the logistic loss. In our experiments we set $t = 2$.

**Group-DRO:**   We run group-DRO (Sagawa et al., 2020, Algorithm 1) with the logistic loss. We set adversarial step-size $\eta_q = 0.05$ which was the best hyperparameter identified by Wang et al. (2022).

