# OpenReview forum: "Undersampling is a Minimax Optimal Robustness Intervention in Nonparametric Classification"
_TMLR — Accepted by TMLR_

### Review · Reviewer_pxEG · 2023-04-11

**Summary Of Contributions:**

This paper contributes to showing minimax lower bounds for imbalanced binary classification by considering two scenarios: label shift and group-covariate shift. The authors aim to convince that the classical approach of undersampling is a reasonable one even though it would not leverage some information from the majority samples, unlike more sophisticated methods such as oversampling and group distributionally robust learning.

For the label shift scenario, the training distribution can have an imbalanced proportion of positive and negative samples. The authors show that the minimax lower bound of the classification risk under this scenario is in the order of $n\_\mathrm{min}^{-1/3}$. This result is notable because the minimax rate is completely free from the majority sample size and captures the trend as the label proportion becomes more extreme. Indeed, the previous minimax bounds in transfer learning and cost-sensitive learning fail to capture this mode appropriately.

For the group-covariate shift scenario, the training samples have another attribute representing two groups in addition to their labels, and we suppose the covariate shift between the two groups. In this case, the authors show that the minimax lower bound is in the order of $n\_\mathrm{min}^{-1/3}$ when the two covariate distributions rarely overlap, and the minimax error becomes closer to the order of $1/n$ when they highly overlap. Thus, the two modes are interpolated smoothly in theory.

Further, the authors unveil that the binning estimator using the undersampled data achieves the obtained minimax rate for both scenarios. With neural network experiments, they additionally confirm that adding the minority samples is more effective in boosting the classification performance than adding the majority samples. These results altogether support the efficacy of the undersampling strategy and the fundamental hardness of learning under the limited availability of the minority samples.

**Audience:**

Yes

**Broader Impact Concerns:**

Since this work is mainly theoretical, there is no concern about broader impacts.

**Claims And Evidence:**

Yes

**Requested Changes:**

Major suggestions are stated as the weaknesses above.

Some minor comments:

- In Section 3.1 (the second paragraph), the classifier domain should be $f: [0,1] \\to \\{-1, 1\\}$, not $f: \\mathbb{R} \\to \\{-1,1\\}$.
- Right before Theorem 4.2, you may not need "the absolute constant c > 0" because it does not appear in the theorem.
- Here and there, I found many typos like "Eq. equation X." For example, you can find them in the statements of Theorems 5.1 and 5.2.
- Right before Lemma B.1, it looks better to finish the sentence such as "the minimax excess risk is lower bounded as follows."

**Strengths And Weaknesses:**

### Strengths

- **Importance of the hardness results**: Although the class imbalance problem has been extensively studied experimentally and theoretically, the fundamental hardness results were lacking. This work shows what we can do under given imbalanced conditions at best. We eventually admit the effectiveness of the undersampling strategy, which is simple but often dismissed, perhaps because it discards some information about the training samples.
- **Educational proof**: The minimax lower bounds in this paper are shown by combining Le Cam's method and the total variation bound. The proof follows the classical flow with a new construction of the hard instance of the training distribution specifically for imbalanced learning. It is written in a transparent way and quite educational.
- **Going beyond the existing analyses**: I like this paper particularly for giving a meaningful extrapolation for the limit $\\rho \\to 0$ (extreme imbalance) in Theorem 4.1 and interpolation for $\\tau \\to \\{0, 1\\}$ (hard/easy transfer) in Theorem 4.2. In particular, the interpolation of Theorem 4.2 gives us insights into when the minority sample size is the performance bottleneck. None of the existing work should be able to provide these insights, including the analyses of transfer learning and cost-sensitive learning.

### Weaknesses

- **Limited analysis for $d=1$**: The analyses in this paper focus merely on the one-dimensional case. Although the authors mention that the extension to the multi-dimensional case is elementary, I do not immediately see how to extend the construction of the "hard" distributions using the hat function beyond the one-dimensional case. It is illustrative to give readers instructions on constructing it when $d>2$. We do not necessarily need the full detail of the construction. In addition, the binning estimator in Section 5 seems to suffer from the curse of dimension under the extreme high-dimensional case. I understand that the authors do not intend to claim that the binning estimator is a "silver bullet," but it should sound fair if you can discuss what happens under the high-dimensional case.
- **Discussion of Kpotufe & Martinet (2018)**: In the last part of Section 4.2, the authors discuss the applicability of the minimax analysis of Kpotufe & Martinet (2018) in the current problem setting, but the discussion seems to be limited to only the zero-overlap case $\\tau=0$. What if $\tau>0$? Can their analysis be applied to imbalanced learning, and if so, is there any agreement/contradiction with your results obtained in Theorem 4.2? Because the framework of Kpotufe & Martinet (2018) is slightly intricate, it is instructive for readers if you have some discussions here. (For the comparison of Theorem 4.1 and the existing analysis of cost-sensitive learning, I could imagine the difference)

---

> ### Author Response · Authors · 2023-04-11
> **Addressed major and minor suggestions.**
>
> Thank you for your careful reading of our paper and for the review.
>
> We have uploaded a revision of the paper with the following changes:
> - We have extended our discussion of the results of Kpotufe & Martinet (2018) in the high overlap case (tau = 1) in Section 4.2.
> - We have added a detailed discussion about extensions to high dimensions in the discussion (Section 7).
> - Fixed the typos and implemented the minor suggestions identified in your review.
>
> Happy to address any additional suggestions or concerns.

---

> > ### Comment · Reviewer_pxEG · 2023-05-17
> > **Reply**
> >
> > I would like to appreciate the authors for providing responses and updating the manuscript.
> > Regarding the extension to the high-dimensional case, now I feel that my initial comment (the binning estimator may suffer from the curse of dimensionality) does not make much sense because the minimax lower bound should scale with the same dependency on the dimensionality, as the authors responded.
> > However, the current proof idea in the updated Section 7 seems to be challenging to verify its correctness.
> > If the authors really want to convince readers of the possibility of the high-dimensional extension, I would recommend writing the sketch more thoroughly.
> > Otherwise, I feel it is better omitted. I do not mind the lack of the high-dimensional extension since I acknowledge the contributions in the initial version.

---

### Review · Reviewer_8Fey · 2023-04-22

**Summary Of Contributions:**

This paper studies a distribution shift setting where the distribution for training is a mixture of a minority distribution and a majority distribution, and the distribution for testing is a uniform average of the minority distribution and the majority distribution. The paper considers two kinds of distribution shift: a label shift and a group-covariate shift. For these two problems, the paper derives lower bounds on the minimax excess risk of order $n^{1/3}_{min}$, where $n_{min}$ is the number of examples for the minority distribution. The paper also shows that the undersampling scheme is able to achieve this minimax optimal rate. Empirical results are also presented.

**Audience:**

Yes

**Broader Impact Concerns:**

No concerns on the ethical impacts.

**Claims And Evidence:**

Yes

**Requested Changes:**

Can we extend the analysis to the general case $d>1$. What is the potential challenge in this extension? Can the subsampling still achieve this minimal optimal rate?

What will happen if we consider a general test distribution, i.e., the test distribution is not a uniform average of the $P_{min}$ and $P_{maj}$?

Can we improve the bound in Thm 4.1 by incorporating the information on the overlap between $P_{min}$ and $P_{maj}$?

**Strengths And Weaknesses:**

Strength:

The paper establishes lower bounds on the minimax excess risk, which shows that the rate depends only on the number of training examples from the minority distribution. The paper shows that a binning estimator applied to undersampling can achieve this minimax rate.

The paper presents experimental results to show that the number of negative examples plays a large role in the performance: increasing only the number of minority examples can improve the performance, while increasing only the number of majority examples does not. This matches well with the experimental results.

Weakness:

The paper only considers the special case $d=1$. This is restrictive since $d$ can be large in practice. Furthermore, the effect of $d$ can be important on the performance. It would be very interesting to develop lower bounds for the minimax rate reflecting the dependency of $d$. As far as I see, the binning estimator may suffer from the curse of dimensionality, and therefore is not a good estimator if $d$ is large.

The paper considers a setting where the distribution for testing is a uniform average of $P_{min}$ and $P_{maj}$. It is not clear to me why the paper considers this specific setting. What will happen if the distribution for testing is another weighted average of $P_{min}$ and $P_{maj}$.

The lower bound in Thm 4.1 does not depend on the overlap between $P_{maj}$ and $P_{min}$. This is different from the bound in Thm 4.2. Can we use the information on the overlap to further improve the lower bound in Thm 4.1

---

### Review · Reviewer_ipms · 2023-04-23

**Summary Of Contributions:**

The paper presents an analysis of binary classification under group imbalance (majority vs minority), when the group membership of individual examples is known. The authors prove lower bounds on minimax excess risk when there is either label shift (there are more examples of one class than another), or group-covariate shift (minority and majority groups have different distributions over the input features x, with varying overlap between these distributions). They find that these lower bounds depend purely on the number of minority class examples. Next, they study a predictor that first undersamples the majority group, then bins the features (scalars in a bounded interval), and then makes a prediction based on majority class of the examples in this bin. Finally, the authors run some simple experiments with neural networks, showing that the accuracy of the learned predictors increases with the increase of the minority samples, and not the majority, as predicted by the theory.

**Audience:**

Yes

**Claims And Evidence:**

Yes

**Requested Changes:**

Comments and questions:

- the authors mention that their findings can be easily extended to a high dimensional feature setting. However, it seems like the undersampling and binning algorithm would have terrible dependence on the dimension, and in practice have no relevance. More generally, I find Section 5 to be almost too toy to be interesting. Due to dimensionality dependence in the most straightforward extension of the results, they don’t really shed light on any realistic empirical studies.
 - Is there a version of undersampling such that the undersampled binning estimator would match the lower bound rate when there is overlap in the group-covariate shift distributions? Perhaps the undersampling should not be done as severely?
- How does the literature on memorization relate to the results presented here? See, e.g., Feldman et al papers on memorization from ~2020, 2021.
- Experiments: it would be natural to also show the results with undersampling. It would also help to further disentangle the contribution to accuracy coming from the majority group examples, and also at least somewhat tie back into the theory section on undersampling.
- I see “robustness” mentioned several times (even in the title), however, it is never properly defined or being directly studied.  Robustness can mean a lot of things and is used differently in various research problems. In my opinion, if used, it should be more formally defined.

Typos and minor comments:

 - “Eq. Equation XX” , see, e.g., theorems 5.1 and 5.2. (\cref error).
 - The sentence just above Section 7 “When we add samples to both…” is incredibly hard to follow, and seems to be logically broken.

**Strengths And Weaknesses:**

Overall, I found the results in the paper interesting and think that the community will see these findings as a valuable contribution. As motivated by the authors, under group imbalance, empirically it is hard to improve upon learning on an undersampled balanced datasets. This has been observed in the empirical work, and is sometimes discussed at an intuitive level. Some results in transfer learning are applicable here too (which the authors mention and discuss relative to their work), but I have not seen work establishing minimax excess risk lower bounds in this particular setting.

The weaknesses of the paper relate to high dimensional extensions, experiments that do not evaluate undersampling,  potentially missed connection to memorization literature. For details, please see the comments and questions in the section below.

---

### Decision · Action_Editors · 2023-05-28

**Recommendation:** Accept with minor revision

**Comment:**

The results are of interest and are mostly well substantiated. However, as the discussion on higher dimensions is not sufficiently well founded, the authors are requested to revise it so that it does not make unsubstantiated claims. Instead, the authors may wish to point to a promising direction without making strong claims. The authors are furhter requested to correct all other minor comments mentioned in the reviews.

**Audience:**

The results explain the phenomenon observed in experiments, which show that undersampling is hard to beat using other seemingly more elaborate strategies, and that adding minority examples is helpful, while adding majority examples is not. These findings will be of interest to many in the community.

**Claims And Evidence:**

This paper shows that in the distribution shift setting with a minority class, there are lower bounds on the minimax excess that depend only on the number of minority samples. The claims are well substantiated, except for the extension of the result to higher dimensions. The proof sketch provided by the authors is not sufficiently convincing, and so it is recommended that the authors remove this discussion.

---

> ### Author Response · Authors · 2023-06-12
> **Thank you**
>
> Thank you! We have made the requested changes and have uploaded a camera ready revision.